# The *fru* gene specifies male cooperative behaviors in honeybee colonies

Sven Köhnen [1] ✉, Pia Ulbricht [1], Alina Sturm [1], Julie Carcaud[2],
Jean-Christophe Sandoz[2], Thomas Eltz [3] & Martin Beye [1] ✉

Conspecific individuals can benefit from behavioral interactions, but whether cooperative behaviors require dedicated control in the nervous system is poorly understood. We examine the genetics underlying obligate cooperative male behaviors in honeybee colonies (*Apis mellifera*) that are deeply hard-wired. We screened for transcription factor genes and found that only the *fruitless* (*fru*) BTB zinc-finger gene was restricted to expression in males and in the nervous system. Reporter coexpression revealed Fru-positive neuronal cell populations that are involved in processing and integrating sensory information. Computer-assisted behavioral tracking of male bees with loss-of-function mutations in experimental colonies revealed that *fru* is specifically required for the rate and/or duration of social approach and feeding behaviors, suggesting that the gene scales bees' participation in the collective nutrition task. Together, our study establishes a gene-based specification of behavior initiation and sustainment and provides insight into the connection between cooperation and defined neural populations.

Behaviors can bind individuals into conspecific groups. In primates and birds, individuals can benefit from cooperative behavioral interactions, as these interactions increase the ability to find food, protect against predators and promote breeding success[1,2]. In the Western honeybee (*Apis mellifera*), a classical example of an organism displaying cooperative behaviors, these interactions are innate, not learned and obligate. They are hardwired into the nervous system, making cooperative behavior accessible to genetic and neural manipulations. The behaviors of honeybee colonies are so well controlled that the individuals function together like a single organism or 'superorganism' that feeds and sustains itself[3,4]. This is the basis of the great general ecological success of social insects[5–7].

The members of the honeybee colony are not homogenous. A colony that is ready to reproduce consists of hundreds of males, a queen and thousands of usually sterile female worker bees. The cooperative behaviors of males inside a colony are of special interest because they support the reproductive success of a colony. Males do not behaviorally compete with other males over females or food

resources in the protected shelter of the colony[8,9]. Instead, males receive nutrition-rich, predigested food from worker bees during their maturation, which involves a sequence of antennation, begging and trophallaxis behaviors to transfer the food[10–12]. The trophallaxis behavior plays a central role in the food exchange among colony members, the nutrition of larvae, adult males and queens and the communication about the food status of the colony[10]. The queen's behavior involves mating with males in the air outside the colony and the laying of eggs for the remainder of her life. The worker bees instead collect pollen and nectar outside the nest and store them in wax-constructed combs inside the colony. The colonies reproduce asexually via the splitting of colony members (swarming).

How the male-specific cooperative behaviors are molecularly controlled in the nervous system is largely unknown. The male-specific anatomy of the nervous system that can control male-specific behaviors is well documented, but not at the level of neural circuitry. Male identity manifests in male-specific odorant receptor gene expression, male-specific number and anatomy of glomeruli in the primary

[1]Institute of Evolutionary Genetics, Heinrich-Heine University, Dusseldorf, Germany. [2]Evolution, Genomes, Behaviour and Ecology, Université Paris-Saclay, Gif-sur-Yvette, France. [3]Department of Animal Ecology, Evolution and Biodiversity, Ruhr University, Bochum, Germany. ✉e-mail: svenkoehnen@outlook.de; martin.beye@hhu.de

olfactory processing center (the antennal lobe (AL)), a male-specific smaller mushroom body (MB) harboring 12% fewer neurons and in male-specific larger optical lobes[13–18]. Compared with worker bees, male-specific neural processing has been detected in the glomeruli of the AL for the queen's sexual pheromone 9-oxo-(E)−2-decenoic acid (9-ODA) using calcium (Ca) imaging[18].

Honeybee males newly evolved their cooperative behaviors, which they display inside the colony, as eusociality evolved during the last 60 million years[19,20]. However, the evolutionary mechanism that sparked the transition to cooperation is lacking[21–24]. Whether the control of the many social interactions inside the densely populated colony indeed requires greater neural processing capacity (or even other cognitive abilities; the "social brain hypothesis") is still under debate, not only for social insects but also for primates[25–28].

A point of entry to understand the control of within-colony cooperative behaviors involves genes that hardwire the male-specific nervous system. If such key genes exist, they must be regulated by the well-characterized sex determination cascade (Fig. 1a)[29–32]. The same versus different complementary sex determiner (Csd) protein variants (expressed from hemizygous or heterozygous csd genotypes) mediate

male or female splicing of the feminizer (fem) transcripts, resulting in Fem protein expression only in females (Fig. 1a)[30,31]. This sex-specific Fem protein activity regulates sex-specific expression of downstream transcription factor genes (TFs) via alternative sex-specific splicing. This involves the conserved doublesex (dsx) and lineage-restricted glubschauge (glu) genes, both of which regulate different aspects of sexual differentiation[29,32]. However, which TF genes specify the nervous system of males underlying the control of their cooperative behaviors is currently unknown.

Hence, the key questions concerning the control of cooperative behaviors are manifold: How many key genes coordinate the hardwiring in the nervous system? What aspects of cooperative behavioral control are genetically specified in the nervous system? What neural processing abilities are involved? What evolutionary mechanisms underlie the origin of cooperative behaviors?

Thus far, we do not know which genes took on the function of hardwiring the male's cooperative behaviors in the nervous system. We thus screened for male-specific spliced transcripts in the nervous system[33] and found that, in addition to previously known glu and dsx only the fruitless (fru, LOC409022) TF gene has male-specifically

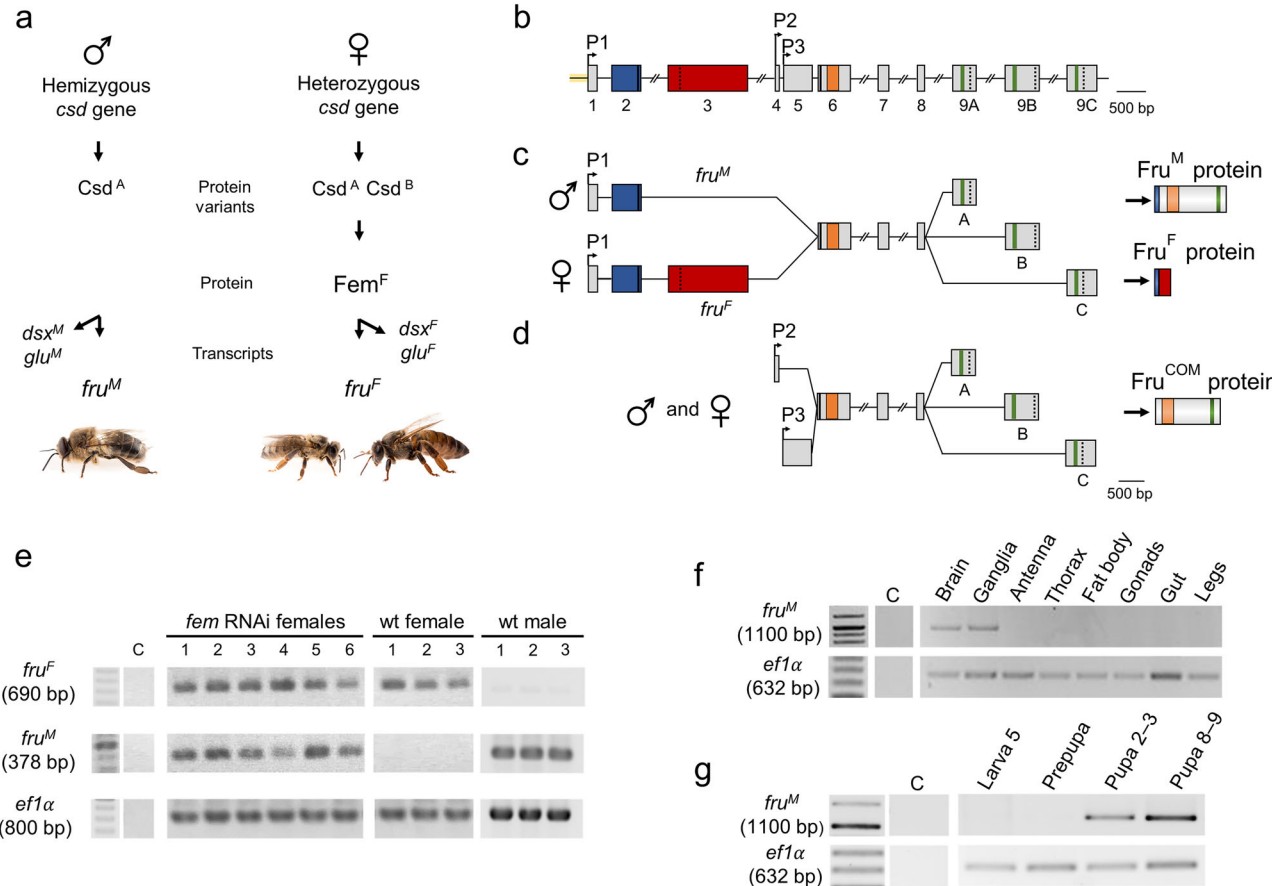

**Fig. 1 | fruitless P1 (fruMP1) transcripts of the male honeybee. a** The sex determination pathway regulates sex-specific alternative splicing of fru transcripts. csd: complementary sex determiner gene that encodes Csd protein variants from different alleles (A, B). Fem: Feminizer protein. dsx: doublesex; glu: glubschauge transcripts. M and F: male- and female-specific variants derived from the splicing process. **b** Genomic organization of the fru gene. P1, P2 and P3 are alternative promoters. Boxes indicate exons. A, B and C are alternatively used exons. Green: zinc finger domains; orange: BTB domain; vertical lines: translational start; vertical dashed line: translational stop. Male-specific parts of the transcript and protein are shown in blue, and the female parts are shown in red. **c** Male- and female-specific fruP1 transcripts. In males the expressed proteins have 386 to 459-amino acids. Note, that if a female protein is produced it will only have 34 amino acids. **d** P2- and P3-derived transcripts express proteins that are common to both sexes (Fru^com). **e** fruP1 transcript splicing in single female larvae after RNAi-induced fem knockdown in syncytial embryos. ef1α (elongation factor 1α) transcripts were used as a reference. wt: wild type. The numbers represent different larvae. **f** Nervous system-specific expression of the fruM transcripts. RT–PCRs were size resolved and semi-quantitatively adjusted using ef1α transcripts as a reference (as in (g)). Three biological replicates in three technical replicates were run for each condition (as in (g)). **g** Developmental onset of fruM transcript expression in the head at the pupal stage. Numbers above gel lanes represent the larval and pupal stages. Please note that the ef1α RT-PCRs in (e) versus (f, g) generated different amplicon sizes because different oligonucleotide primer pairs were used. C is negative control for PCR. Source data of the gel pictures are provided as a Source data file.

spliced transcripts (Supplementary Table 1). Because this Fru BTB zinc finger transcription factor is expressed only in the male nervous system and its homolog in *Drosophila melanogaster* is involved in hardwiring sexual behaviors[34,35], we set out to examine the function of this gene product in manifesting the male's cooperative behaviors in honeybee colonies.

## Results

### The male-specific Fru^M protein is highly spatially and temporally restricted in the nervous system

Our detailed nucleotide sequence transcript analyses revealed that the *fru* gene in honeybees is transcribed from at least three promoters based on three different transcription start sites. The three transcripts that derive from promoters P1 to P3 encode protein variants with a common Broad-Complex, tramtrack and Bric-à-brac (BTB) domain and three alternative zinc finger (ZnF) domains that are produced via alternative splicing and inclusion of either exons 9A, 9B and 9C (Fig. 1b–d; EMBL-EBI accession number: OZ362630-OZ362641). Only the P1-derived *fru* transcripts (*fruP1*) are sex-specifically spliced. In males, the skipping of exon 3 results in male-specific *fruP1* transcripts (*fru^M*), which encode 386- to 459-amino-acid-long protein isoforms with three alternative ZnF domains, suggesting that this is a male-specific BTB ZnF transcript, *fru^M* (Fig. 1c). In females, exon 3 is retained, which results in an early translation stop codon of female-specific *fruP1* transcripts (*fru^f*) and a predicted 34-amino-acid-long protein that lacks the BTB and ZnF domains (Fig. 1c). To determine whether sex-specific *fruP1* transcripts are controlled by the sex determination cascade, we performed RNAi-induced knockdown of the *fem* gene, a key sex determination regulator[31] (Fig. 1a). The *fem* RNAi-treated females produced male-specific *fruP1* transcripts but also female-specific transcripts at larval stage 4 (Fig. 1e and Supplementary Table 2) suggesting a partial shift toward male splicing in response to temporally restricted *fem* knockdown, which we induced in the embryo[31]. This result suggests that this male-specific splice regulation requires the absence of female-determining activity from the *fem* gene, a component of the canonical sex determination pathway[30,31,36].

In newly eclosed adult males, sizeable amounts of *fru^M* transcripts are found in the brain and ganglia but not in other tissues, including the thorax muscle, fat body, or gonads (Fig. 1f). This finding suggests a spatial restriction of *fru^M* transcripts to the nervous system. During development, sizeable quantity of *fru^M* transcripts are found from early pupation stages on in the head (Fig. 1g). This finding indicates that expression starts in the nervous system at early pupation, when the adult nervous system is formed[37]. Next, we developed an antibody against a sequence of the BTB domain. Using confocal light microscopy sections, we found that the anti-Fru antibody labels male but not female pupal brains in spatially characteristic patterns (Fig. 2a, b and Supplementary Movies 1 and 2), suggesting that these male-specific labels derived from P1, but not P2 or P2 transcripts. To examine whether this male-specific expression pattern was specifically derived from Fru^M protein expression, we generated *fruP1^-* mutants. We deleted the P1 promoter and exons 1 and 2 sequences using the CRISPR/Cas9 method in embryos[29,32,38] (Fig. 2c), reared homozygous queens carrying this deletion, and confirmed the *fruP1^-/-* genotype at the nucleotide sequence level (Supplementary Fig. S1). These queens produced haploid unfertilized eggs from which our *fruP1^-* male progeny were derived. The *fruP1^-* males lacked the *fru^M* transcript (Supplementary Fig. S2) and did not present consistent male-specific Fru protein labels as observed in staining of the wt males (Fig. 2a and Supplementary Movie 3). These results suggest that the male-specific Fru^M protein is expressed from the male *fru^M* transcripts, that the anti-Fru antibody labels Fru protein expression, and that the *fruP1^-* mutant is a loss-of-function mutation of *fru^M*.

We found that our anti-Fru labels colocalized with nuclei (Supplementary Fig. S3). These labels were mostly organized in characteristic clusters, which were frequently located in the outer anterior and posterior sections of the male brain (Fig. 2a, b). We also detected such sparse Fru protein labels in the ganglia (Supplementary Fig. S4), suggesting that Fru^M expression expands to the ventral nerve cord. To determine how many cells express the Fru^M protein, we counted Fru-labeled nuclei in the adult male midbrain (which consists of approximately 400,000 neurons[16]) and found that approximately 1800 cells expressed Fru^M proteins (Fig. 2d). Collectively, these results suggest that the male-specific transcription factor protein Fru^M is highly spatially and temporally regulated in the developing and adult male nervous systems.

### Fru^M-expressing cells are involved in processing and integrating information from different sensory modalities

As it is unclear where Fru^M is expressed in the male brain, we inserted the myrGFP/P2A/T2A coding sequence into the *fru* locus using the CRISPR/Cas9 method[29,38,39] so that the *fruP1* transcripts expressed both membrane-tethered GFP (myrGFP) and Fru^M proteins (Fig. 3a, Supplementary Fig. S5). We examined two neural clusters to determine whether the insertions generated robust labeling of Fru-expressing cells. The number of Fru-expressing cells in the *fru^myrGFP* males did not differ from that in the wt males as revealed from anti-Fru antibody staining, suggesting that this genomic insertion did not disrupt the formation of these neurons (Fig. 3b and Supplementary Fig. S6; Cluster I (P = 0.54), Cluster II (P = 0.33), MWU test). On average, 100% and 96% of the anti-Fru-labeled cells in the two clusters were also positive for GFP, suggesting robust GFP labeling of the Fru-expressing (Fru^M+) cells (Fig. 3C and Supplementary Table S3).

We detected a number of Fru^M+ cell populations in brain areas with known functions[14,40,41] (Fig. 3d and Supplementary Movies 4–7). We observed Fru^M+ neurons in periesophageal neuropils (PPs) in the posterior region (Fig. 3d), which are involved in processing mechanosensory and gustatory information. One tract runs medially on each side of the brain, possibly connecting with the superior medial protocerebrum (SMP, Fig. 3d). In the antennal nerve, we identified Fru^M+ neurons (antennal tracts T5 & T6, Supplementary Movies 4 and 5) that project into the antennal mechanosensory and motor center (AMMC; Supplementary Movies 4 and 5) and possibly into the subesophageal zone (SEZ; Fig. 3d), indicating that mechanosensory and/or gustatory neurons are involved. The olfactory sensory neurons (OSNs) entering the antennal lobe (AL) were not stained (Supplementary Movies 4 and 5). Many glomeruli in the AL (the primary olfactory processing center) exhibit Fru^M+ in their core region. This includes the male-specific macroglomeruli MG1 and MG2 (AL, Fig. 3d), the latter of which processes the sensory input from the queen pheromone 9-ODA[18]. The glomerulus core Fru^M+ circuitry primarily contains processes from local neurons (LNs), because projection neurons (PNs) extending to the mushroom bodies (MBs) were not labeled (Supplementary Movies 4 and 4). The lateral horn, another center of the olfactory pathway, shows some Fru^M+ fibers (LH, Fig. 3d). Several Fru^M+ neurons in the middle brain belong to output tracts of the visual system. The anterior optic tract (AOT) has some Fru^M+ cells that project to the anterior optic tubercle (AOTu, Fig. 3d, Supplementary Movies 2–7). Three optic commissures present Fru^M+ cells that connect the two optic lobes (inferior and posterior optic commissures, IOC and POC) and the two AOTu (vTUTUT: ventral tubercle-tubercle tract; Supplementary Movies 4 to 7). Fru^M+ neurons are also observed in the optic lobes with diffuse layers in the medulla, clearly stained fibers in the inner chiasma, and three distinct layers in the lobula (possibly corresponding to layers 1, 3, and 5)[42] (Supplementary Movie 8 and 9). In the higher-order processing center, the mushroom body (MB), we detected Fru^M+ neurons in the calyx (see medial calyx (MC), Fig. 3d). Fru^M+ neurons were found in the lips (li, which receive projections from the antennal lobe[43]), in the outer collar zone (co, which receives visual input from the lobula and medulla[44]) and in the basal ring (br, which

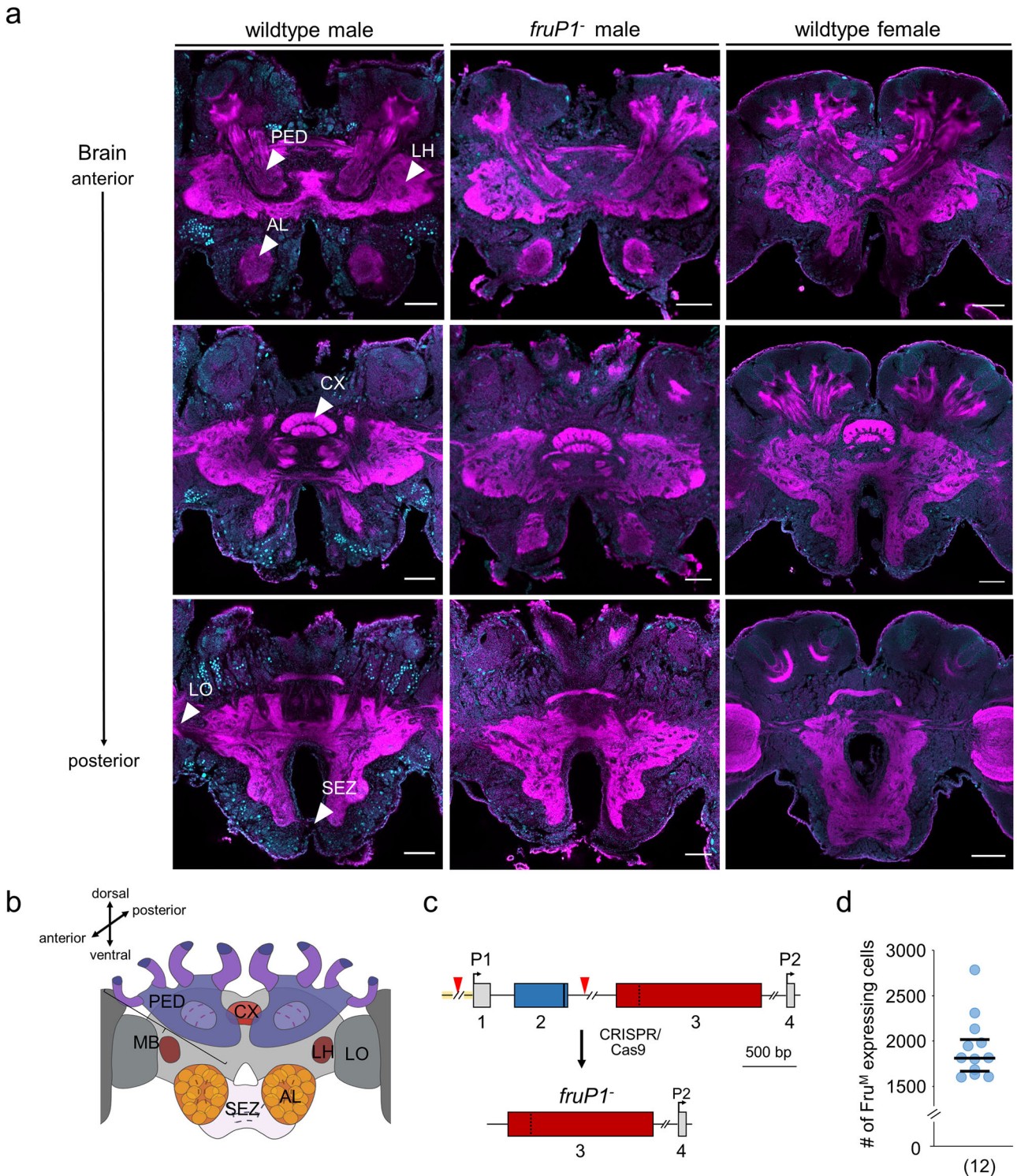

**Fig. 2 | Highly spatially restricted expression of the FruM protein in the nervous system. a** Anti-Fru immunostaining of midbrains of wt males, wt female workers and *fruP1-* pupae (P5 stage). Cyan: anti-Fru staining. Magenta: phalloidin counterstaining. *n* = 7 bees were examined. Scale: 100 μm. **b** Scheme of the anatomy of the bee brain. MB, mushroom body. LO, lobula. LH, lateral horn. PE, peduncle. CX, central complex. AL, antennal lobe. SEZ, subesophageal zone. This graphic was created using BioRender. **c** Deletion of the P1 promoter and exon 1 and 2 genomic sequence generated *fruP1-* males. The boxes represent exons. Blue: male specific. Red: female specific. Arrowheads: target sites of sgRNAs 6 and 12. **d** Number of FruM-expressing cells in the adult male midbrain. Twelve brains were examined. Medians (middle line) and quartiles are shown.

receives multisensory input from at least the visual and olfactory systems[45]; Fig. 3d). Intrinsic MB neurons, including a subpopulation of type I Kenyon cells (KCs) and some type II KCs, are Fru positive. The neurites of type I KCs extend along the peduncle and bifurcate into the medial and vertical lobes of the MB, whereas those of type II KCs

project only into the most ventral part of the vertical lobe (VL, Fig. 3d). This manifests as different layers in the VL, which include from rostral to caudal, type I KC input from the basal ring (Supplementary Movies 4 and 5), the collar, the lip (VL, Fig. 3d) and type II KC input (Supplementary Movies 4 and 5). FruM+ neurons are also prominently

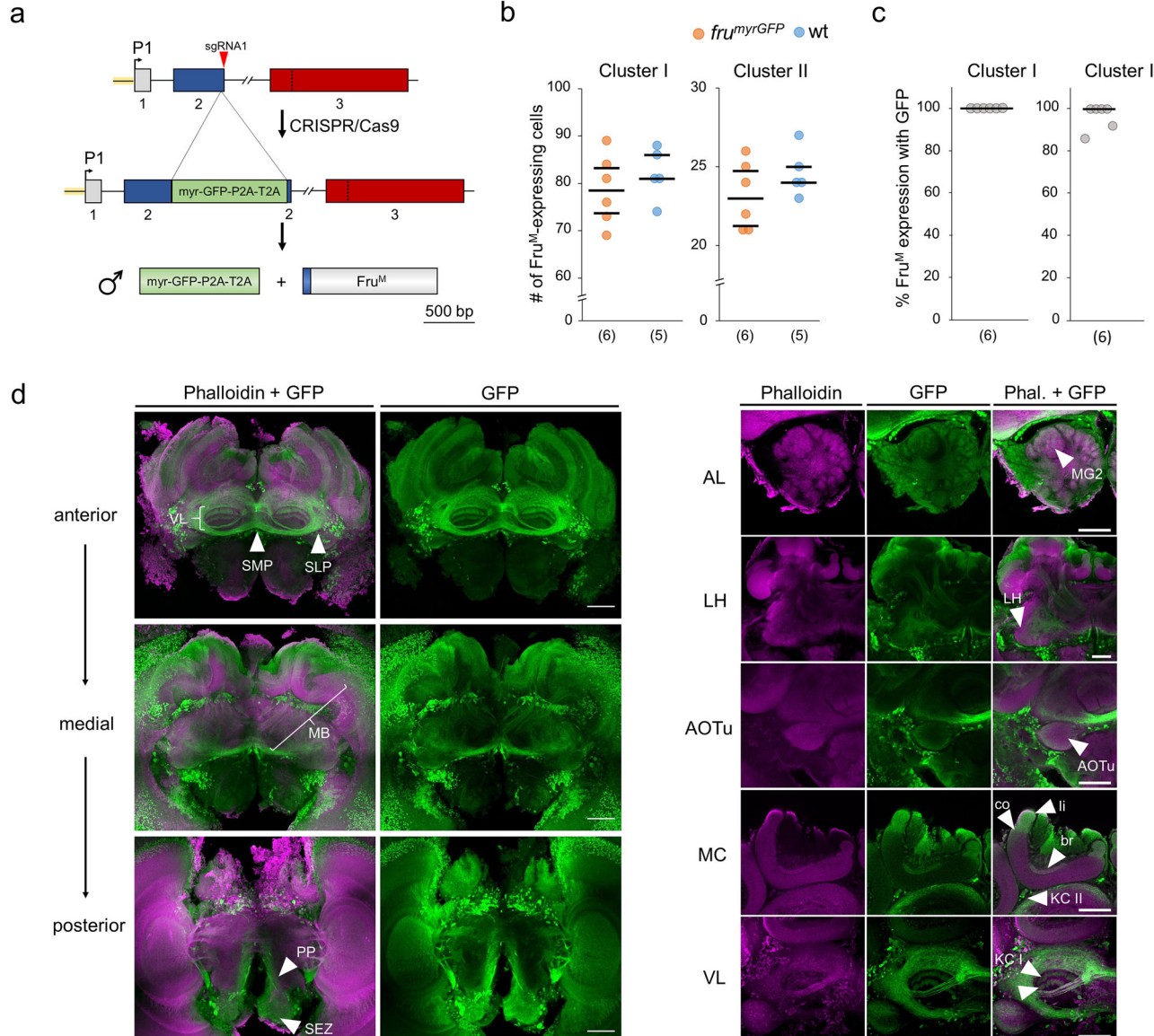

**Fig. 3 | Projection patterns of Fru^M-expressing (Fru^M+) cells in the midbrains of males. a** Insertion of a myrGFP coding sequence into the *fru* genomic locus. P1: promoter 1. Green box: donor sequence encoding the membrane-tethered myrGFP protein and the P2A/T2A peptides mediating the two proteins. **b** Fru^M-expressing (Fru^M+) neuronal somata in Cluster I (approximately 78 cells) and Cluster II (approximately 23 cells) of *fru^myrGFP* and wt males. Cluster I ($P = 0.54$, $z = 0.74$); Cluster II ($P = 0.33$, $z = 1.0$), MWU test. **c** Proportion of Fru^M cells expressing myrGFPs in the somata of the two clusters. All statistical tests were two-sided. (n) below graphs (**b**) and (**c**): the number of bees examined. **d** Confocal images of the *fru^myrGFP* male brain. Labeling of F-actin (via phalloidin, magenta) and Fru^M (via anti-GFP, green) proteins. *n* = 8 bees were examined. **Left panel:** Anterior, medial and posterior overviews of the Fru^M+ circuitry in the midbrain. Anterior: Fru^M+ axons in

the vertical lobe (VL), superior medial protocerebrum (SMP, arrow), and superior lateral protocerebrum (SLP, arrow). Medial: Fru^M expression in the peduncle and calyces of the mushroom body (MB). Posterior: Neuronal cells in the periesophageal neuropil (PP, arrow) and subesophageal zone (SEZ, arrow) showing Fru^M expression. **Right panel:** Details of the Fru^M+ circuitry. AL: antennal lobe. Fru^M+ in macroglomerulus 2 (MG2; arrowhead). LH: lateral horn. Fru^M+ in the lateral horn (arrowhead). AOTu: anterior optical tubercle. Fru^M+ cells of the optical tubercle (arrowhead). MC: medial calyx. Fru^M+ in the basal ring (br, arrowhead), collar (co, arrowhead), lip (li, arrowhead), and Fru^M+ somata of type II Kenyon cells (KC II, arrowhead). VL: vertical lobe. Fru^M+ type II Kenyon cells in the strata of the vertical lobe that project from the lip (arrowhead) and outer collar zones (arrowhead below). Scale 100 µm.

found in anterior parts of the SMP around the medial lobe and in regions corresponding to the superior lateral protocerebrum (SLP, Fig. 3d), including those that connect the two hemispheres[41] (Fig. 3d), which possibly represent a hub for preprocessing input from the MB and possibly other regions of the brain. Collectively, these results suggest that the Fru^M+ circuitry is involved in the processing of sensory information (olfaction, vision, and possibly gustation and mechanosensation) from the periphery to higher-order processing centers such as the mushroom bodies.

## Fru^M+ circuitry has a male-specific anatomy

To understand whether the Fru^M+ circuitry has a male-specific anatomy that can hardwire male-specific behaviors, we generated *fruP1^myrGFP* female workers and compared the labeled structures with those of *fruP1^myrGFP* males. In females, we denote these labeled cells as myrGFP^P1+ because the GFP is generated from the same P1 transcript as in males, but these cells donot express the male Fru^M protein. At the gross level, myrGFP^P1+ and Fru^M+ circuitries in females and males were similar, but neural populations involved in olfactory visual information processing

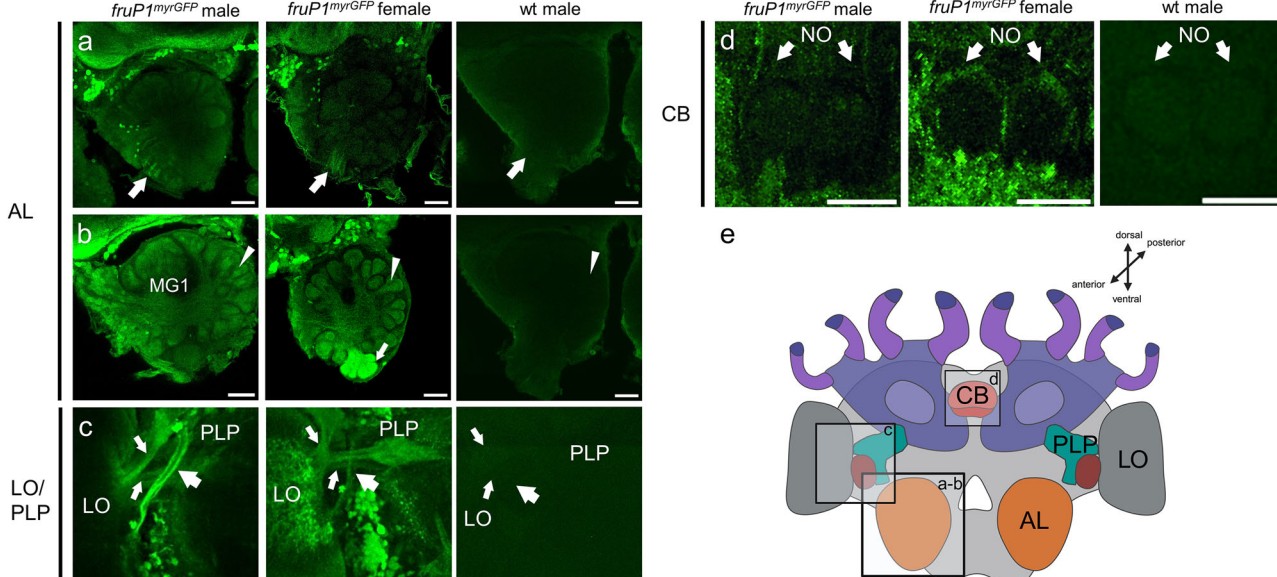

**Fig. 4 | Male-specific identity of the Fru^M+ circuitry. a–d** Dimorphic *fruP1^myrGFP*-labeled structures and circuitry in male and female worker midbrains. Because Fru^M proteins are expressed only in males, P1-derived myrGFP expression labels Fru^M+ cells in males and myrGFP^P1+ cells in females. AL antennal lobe, MG macroglomerulus, LO lobula, PLP posterior lateral protocerebrum, CX central complex, NO noduli. Structures were visualized using anti-GFP antibody. Anti-GFP staining of wt males was used as negative control. **a** The antennal nerve of males and worker females is GFP labeled (arrows). **b** The male-specific macroglomerulus (MG1) has Fru^M+ circuitry in the core. Other glomeruli of males and glomeruli of females have labeling of circuitry in the core (arrowheads). Worker bees have a cluster of ventro-medial glomeruli that have myrGFP^P1+ circuitry not only in the core but also in the cortex (arrow), a pattern not found in male glomeruli. This female-specific labeling of the cortex suggests that OSNs in the antennal nerve (Fig. 4a) are Fru^M+ negative in males and myrGFP^P1+ positive in females. **c** A male-specific Fru^M+ optical tract that connects the LO with the PLP (large arrow). The other two tracts are also labeled in worker bees (small arrows). **d** A layer of myrGFP^P1+ circuitry is found in the NO of female workers. The corresponding Fru^M+ circuitry is absent in males. **e** Schematic overview of the honeybee brain and relevant structures. This Graphic was created using BioRender. *n* = 6 males and *n* = 7 female bees were examined. Abbreviations as above. Scale 50 μm.

showed notable differences (arrows and arrowheads in Fig. 4 and Supplementary Movies 6, 7, 11 and 13). The antennal nerve is labeled in both male and female worker bees (arrow, Fig. 4a and Supplementary Movie 6, 11 and 13). In workers, a small number (approximately a dozen) of glomeruli on the ventro-medial side of the AL had myrGFP^P1+ labels both in the core and cortex (arrow, Fig. 4b), whereas in males Fru^M+ labels were consistently restricted to the core (arrowhead, Fig. 4b). This female-specific labeling of the cortex suggests that OSNs in the antennal nerve are Fru^M+ negative in males but some are myrGFP^P1+ positive in females. This is supported by a more restricted labeling of the nerve in males than in females (arrows, Fig. 4a: note that the antennal nerve contains approximately 4 times more neurons in males than in female workers[14]). Hence, sensory neurons from the antennal nerve that bypass the AL and glomeruli on the ventral side (antennal tracts T5 &T6) towards the AMMC and potentially the sub-esophageal zone (SEZ) were labeled in males but also in female worker bees. The set of worker-specific labeled glomeruli may correspond to the T3b cluster, which may be involved in the processing of cuticular hydrocarbon cues[14,46,47]. Fru^M+ circuitry is present in the core of the macroglomeruli, which are male-specific structures not found in workers (MG1, Fig. 4b and Supplementary Movies 6, 11 and 13). The Fru^M+ circuitry of the male visual system has an extra male-specific tract not found in the myrGFP^P1+ circuitry of workers (large arrow, Fig. 4c and Supplementary Movies 7). This male-specific tract projects from the lobula (LO) to the posterior lateral protocerebrum (PLP). However, two other labeled tracts in the same area are shared between males and worker bees (small arrows, Fig. 4c). In the central complex, a few labeled myrGFP^P1+ fibers were found in the noduli (NO) of female worker bees, but corresponding Fru^M+ fibers in males were absent in that region (arrows, Fig. 4d and Supplementary Movies 6, 11 and 13)[48]. We conclude from these comparisons that the Fru^M+ circuit has a male-specific identity that manifests at the gross level in the anatomy of the antennal nerve, macroglomeruli, and innervation of the core/cortex of the glomeruli and optic tracts.

## The initiation and sustainment of social feeding behavior dysfunction in *fruP1⁻* males

Sexual maturation of males depends on protein-enriched nutrition which the males receive from the worker bees via trophallaxis behavior. This process requires a sequence of behaviors with the option to quit at a particular step. The male approaches and initiates contact with the worker bee's head with its antennae while displaying begging behavior. This behavior eventually leads to trophallaxis behavior, in which the male extends its proboscis to the donor worker bee's mandibles to receive liquid food via mouth-to-mouth transfer[10,11]. The (nurse) worker bees thereby provide honey liquid from the expandable part of their gut ("honey" stomach), which is enriched with proteins from the hypopharyngeal glands[10,12]. The components of these proteins are derived from digested and metabolized pollen, which males can hardly digest[10,12].

To determine whether the male-specific Fru^M transcription factor protein hardwires neural circuits that control the social feeding behaviors, we studied loss-of-function *fruP1⁻* males in small experimental colonies. Twenty *fruP1⁻* males and wt males, together with 460 worker bees and a queen, were assembled on combs (Fig. 5a). These combs contained the same amount of pollen and honey stored at the same location in the cells (Supplementary Fig. S7). The behavior of each male was tracked using unique two-dimensional barcode labels and our computer-based Bee Behavioral Annotation System (BBAS)[49] (Fig. 5a).

We observed that the rate of begging behavior was markedly and significantly reduced by approximately twofold in *fruP1⁻* males compared with wt males (*P* = 0.004, MWU test; Fig. 5b). Moreover, the rate of trophallaxis behavior was reduced by more than twofold on average

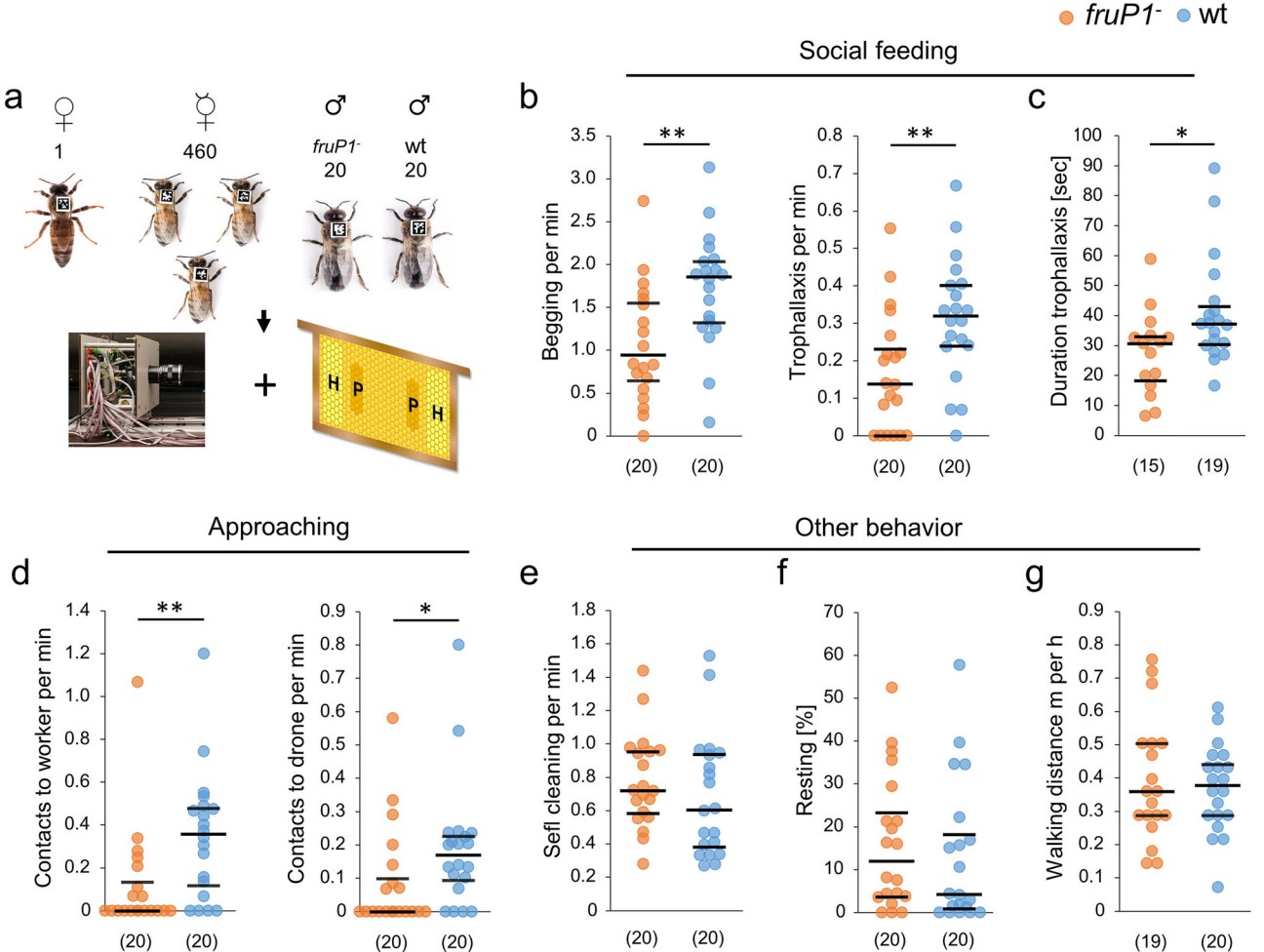

**Fig. 5 | Fru^M protein activity is required for social feeding behaviors in the colony. a** Behavioral examination of single *fruP1⁻* and wt males in small honeybee colonies using two-dimensional barcoding and computer-based tracking. **b** Rates of begging (*P* = 0.004, *z* = 2.8, MWU test) and trophallaxis (*P* = 0.005, z = 2.7, MWU test) behaviors. **c** Duration of trophallaxis behavior (*P* = 0.027, *z* = 2.2, MWU test). **d** Rates of approaching behavior toward worker bees (*P* = 0.004, *z* = 3.2, MWU test) and toward males (*P* = 0.015, *z* = 2.4, MWU test; Fig. 5d; Supplementary Movies 18–21). **e**. Rate of self-cleaning behavior

(*P* = 0.21, *z* = 1.2, MWU test). **f** Proportion of time spent resting (*P* = 0.3, *z* = 1.0, MWU test). **g** Locomotor behavior (*P* = 0.94, *z* = 0.9, MWU test). *fruP1⁻* and wt males were compared in two colony replicates. Medians (middle line) with quartiles Q1 and Q3 are shown. (n) below graph (**b**) to (**g**): the number of males examined. min minutes, sec seconds, m meters, h hours. **P* < 0.05; ***P* < 0.01. All statistical tests were two-sided.

in *fruP1⁻* versus wt males (*P* = 0.005, MWU test; Fig. 5b). We found that the duration of trophallaxis behavior was slightly but significantly reduced in *fruP1⁻* compared with wt males (*P* = 0.027, MWU test; Fig. 5c). However, the movement patterns during the execution of begging and trophallaxis behaviors were stereotypic and unaffected, suggesting that the initiation of begging and trophallaxis and the sustainment of trophallaxis behaviors were specifically impaired in *fruP1⁻* males (Supplementary Movies 14–17).

Begging behavior requires that a male approach another colony member. To understand whether the male's approaching behavior is actively controlled and innately specified via the Fru^M protein or whether it arises passively from random locomotion-driven encounters, we examined the rate at which males approached with their antennae the abdomens and thoraxes of worker bees or males, a behavior that does not lead to begging behavior. We observed that these other social contacts were also substantially reduced irrespective of whether the approached bee was a worker (*P* = 0.004, MWU test) or a male (*P* = 0.015, MWU test; Fig. 5d; Supplementary Movies 18–21). This result suggests that approaching behavior is impaired in *fruP1⁻* males and is innately specified. The rates of other male behaviors, such as self-cleaning behavior (*P* = 0.21, MWU test; Fig. 5e), resting behavior (*P* = 0.3, MWU test; Fig. 5f), and locomotion

behavior (*P* = 0.94, MWU test; Fig. 5g), were not affected in *fruP1⁻* versus wt males, suggesting that social feeding behaviors were particularly impaired (Supplementary Movies 22–25). Collectively, these results suggest that the initiation of approaching, begging and trophallaxis behaviors and the sustainment of trophallaxis were specifically impaired in the *fruP1⁻* males.

**Other male-specific within-colony behaviors are also impaired**
Surprisingly, we observed that *fruP1⁻* males repeatedly entered the pollen-containing cells (average 17 times per hour), whereas wt males never did so (*P* < 0.001, MWU test; Fig. 6a). The *fruP1⁻* and wt males did not enter the honey cells (no difference: *P* = 0.79, MWU test; Fig. 6a), a behavior which has been reported for older males[50]. Our *fruP1⁻* and wt males sporadically entered the empty cells, but the rates did not differ (*P* > 0.96, MWU test; Fig. 6a). The movements of *fruP1⁻* males into pollen cells followed the same stereotypic pattern as those of wt males that entered an empty cell (which often involved bouts of repeated cell entries (Supplementary Table S4 and Supplementary Movies 26–29)). These results suggest that the choice of which cell type is entered is disrupted in *fruP1⁻* males. The increase in the entering of pollen cells cannot be explained by higher visiting rates to the pollen stores because the pollen area visits were lower in *fruP1⁻* than in wt males

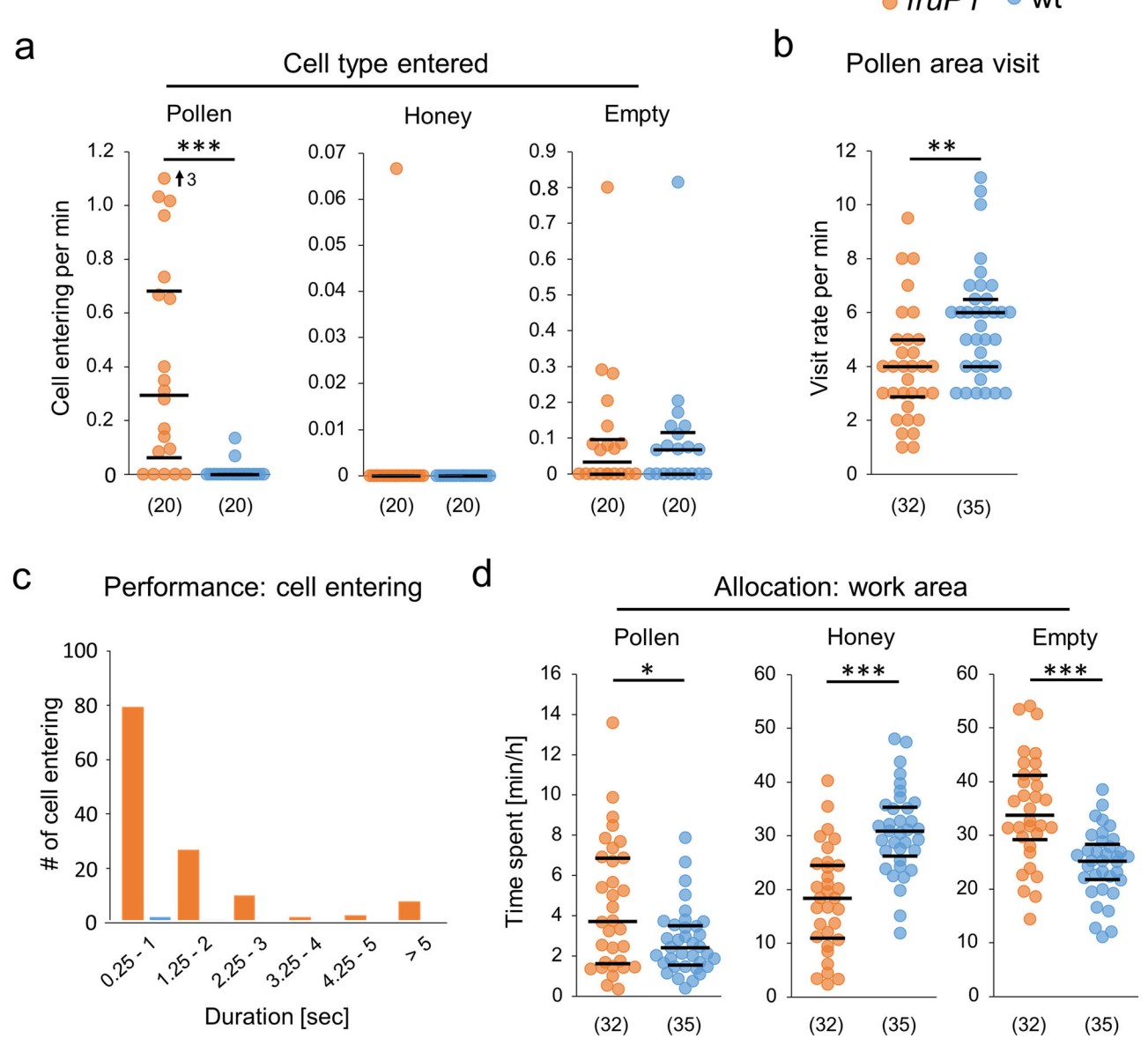

**Fig. 6 | Fru$^M$ protein activity is required for the control of food store-associated behaviors. a** Rates of entering different types of cells on the comb: pollen cells ($P = 1.7 \times 10^{-5}$, $z = 4.3$, MWU test), honey cells ($P = 0.79$, $z = 0.26$, MWU test), and empty cells ($P = 0.94$, $z = 0.07$, MWU test). **b** Rate of visiting pollen areas ($P = 4.2 \times 10^{-6}$, $z = 4.6$, MWU test). **c** Duration of entering the pollen cells of the mutant males. **d** Time spent in the different work areas of the comb: pollen store

($P = 0.045$, $z = 2$, MWU test), honey store ($P = 1.6 \times 10^{-6}$, $z = 4.8$, MWU test), and empty area ($P = 4.1 \times 10^{-5}$, $z = 4.1$, MWU test). Medians (middle line) and quartiles are shown. (n) below graphs (**a**, **b**, **d**): the number of males examined. sec seconds, min minutes, h hours. In **a**, an outlier is marked with an arrow, and the value is provided. *$P < 0.05$; **$P < 0.01$; ***$P < 0.001$. All statistical tests were two-sided.

($P = 0.002$, MWU test; Fig. 6b). Because the entering of pollen cells in *fruP1⁻* males lasted for more than 2 s (Fig. 6c), we investigated whether the *fruP1⁻* males consumed pollen, a food source they can hardly digest[10,12]. No sizeable amount of pollen grains was found in the midguts and hindguts of *fruP1⁻* males (Supplementary Fig. S8), suggesting that the entering of pollen cells do not result in a consumption of pollen.

Males do not perform any behavioral task inside the colony. They also do not compete with other males over food sources or females[8]. Hence, males are usually located on the periphery and in the honey areas rather than in the densely populated center of the colony, in which the brood areas and pollen stores are found[51]. We examined whether the location of the males relative to the food stores is also innately programmed via Fru$^M$. We found that the *fruP1⁻* spent more time in the pollen and empty areas but less time in the honey area than

wt males (Fig. 6d), suggesting that the allocation behaviors towards food stores were impaired in *fruP1⁻* mutants. Together, these findings suggest that the allocation towards the food stores and the choice of which food is visited in the cells are innately programmed and involve Fru$^M$.

**Sex pheromone-related behaviors, but not general sensorimotor functions are impaired in *fruP1⁻* males**

Males mate with queens outside the colony during a mating flight. Virgin queens release a sex pheromone (9-oxo-(E)-2-decenoic acid (9-ODA)), which attracts males during these flights[52,53]. To understand whether the Fru$^M$ protein also contributes to sexual behavior in the nervous system, we examined changes in locomotion behavior in response to the sex pheromone 9-ODA[54] in a behavioral Petri dish assay. Compared with wt males, *fruP1⁻* males exhibited reduced

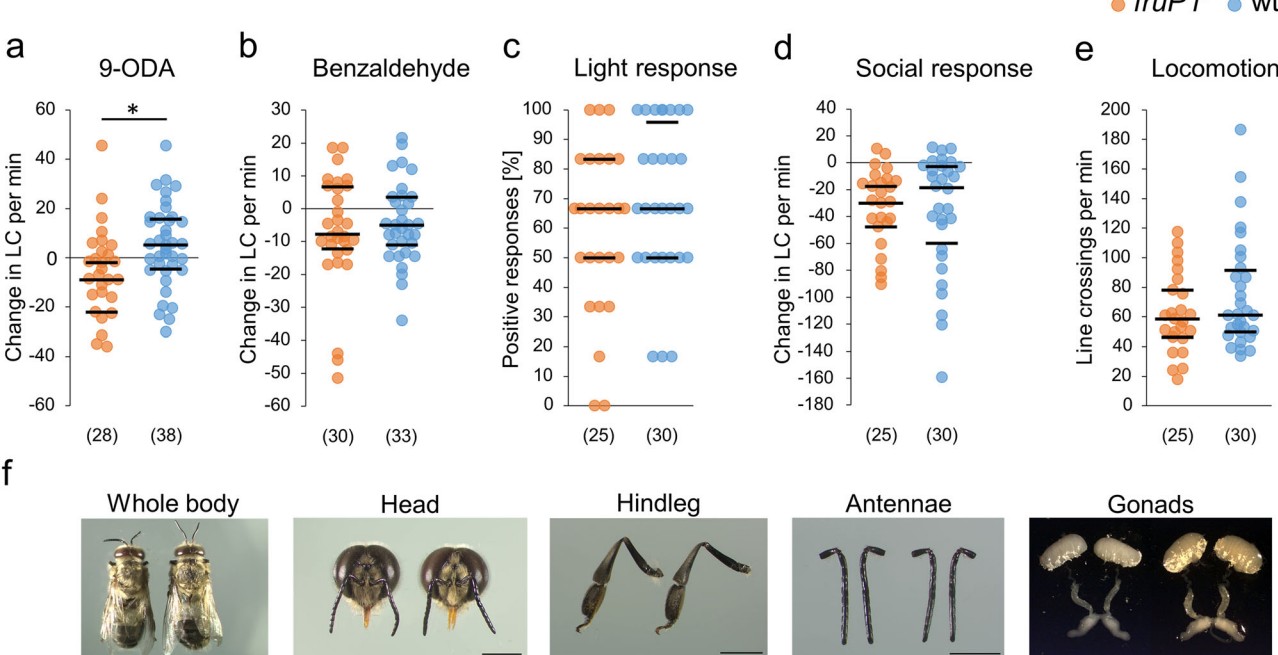

**Fig. 7 | FruM protein activity is required for the sex pheromone response but not for general sensorimotor functions. a** Locomotor changes (line crossing of a grid in an arena) in *fruP1−* and wt males in response to the sex pheromone 9-ODA ($P = 0.01$, $z = 2.5$, MWU test). The males were at least 8 days old. **b** Changes in the locomotion of *fruP1−* and wt males in response to benzaldehyde ($P = 0.84$, $z = 0.2$, MWU test). Both *fruP1−* males and wt males responded to benzaldehyde ($P < 0.05$, $z > 1.9$, Wilcoxon rank sum test against zero). **c** Locomotion response to light stimuli ($P = 0.75$, $z = 0.3$, MWU test). **d** Locomotion changes in response to social contact with two worker bees ($P = 0.66$, $z = 0.4$, MWU test). Both *fruP1−* and wt males responded to these social interactions ($P < 0.001$, $z > 3.9$, Wilcoxon rank sum test against zero). **e** Locomotion without stimulus. $P = 0.3$, $z = 1.0$, MWU test). **f** Morphology and anatomy of wt (left) and *fruP1−* (right) males. At least $n = 7$ bees were examined. Scale: 2 mm. Medians (middle line) and quartiles are shown. (n) below graphs (**a**–**e**): the number of males examined. min: minutes. *$P < 0.05$. All statistical tests were two-sided.

locomotion in response to 9-ODA ($P = 0.011$, MWU test; Fig. 7a), indicating that sexual behavior is also impaired in *fruP1−* males.

To exclude the possibility that the dysfunction of sexual and within-colony behaviors in *fruP1−* males results from a general impairment of sensorimotor functions (such as a general dysfunction of neuronal processing or moving abilities), we quantified behavioral responses to olfactory, gustatory, mechanical, or visual stimuli. We found that the locomotion responses to the repellent benzaldehyde ($P = 0.84$, MWU test; Fig. 7b), light ($P = 0.75$, MWU test; Fig. 7c), and contact with worker bees ($P = 0.66$, MWU test; Fig. 7d) did not differ between *fruP1−* and wt males. The locomotion behaviors of *fruP1−* and wt males in the absence of these stimuli also did not differ ($P = 0.3$, MWU test; Fig. 7e). Both groups also responded with proboscis extension to honey ($P = 0.51$, Fisher's exact test; Supplementary Table 5). Moreover, impaired viability cannot explain the dysfunction of male-specific behaviors because the *fruP1−* mutation did not influence the survival rate of adult males (log-rank test, Chi² = 0.22, df = 1, $P = 0.64$; Supplementary Fig. 9). The behavioral dysfunctions also cannot be explained by a malfunction and malformation of external and internal structures at the gross level because the external head, antennae, abdomen and leg morphology as well as the reproductive organ anatomy of *fruP1−* males had a wt phenotype (Fig. 7f). These results suggest that general sensorimotor functions are intact in *fruP1−* males. Hence, the impairment of general sensorimotor functions cannot explain the dysfunction of male-specific behaviors in the mutants.

Cuticular hydrocarbons (CHCs) can provide important cues for social interactions among colony members[55–57]. To examine whether *fruP1−* mutation altered CHC profiles, we examined the CHC profiles via gas chromatography–mass spectrometry (GC–MS) analysis. We observed that all the compounds were present in *fruP1−* and wt males and that all the compound classes were equally represented (Supplementary Fig. 10). However, there was a significant difference in the overall CHC composition between the mutant and wt males (Supplementary Fig. 10). Next, we examined whether the individual CHC composition profiles of *fruP1−* and wt males produced separate clusters in two-dimensional multidimensional scaling (MDS) space (Supplementary Fig. 11). We found that the individual profiles partially overlapped between the groups, suggesting that the CHC are unlikely to have triggered a behavioral difference between the groups of mutant and wt bees.

## The gross neuropil structure and odor processing features of the AL are not impaired in *fruP1−* males

How can the FruM protein specify these male behaviors? We propose that the FruM protein specifies connectivity or functional features in the nervous system that determine behavioral control. To understand the role of the FruM protein in the programming of behaviors, we examined chemosensory receptor abundance, neuropil anatomy, and odor processing in *fruP1−* males.

We found no changes in the expression of odorant receptor (OR), odorant binding protein (OBP), gustatory receptor (GR), or chemosensory protein (CSP) genes in the antennae of *fruP1−* versus wt males using the RNAseq method (Supplementary Fig. 12; EMBL-EBI accession number: PRJEB100586). This result suggests that the repertoire of chemosensory proteins involved in odor detection is not impaired in *fruP1−* males. This includes the *Or11* gene, which expresses a key odorant receptor protein involved in the detection of the queen pheromone 9-ODA[15], a gene that is highly male-specifically expressed (>25-fold higher expression in males than in worker bees). Our optical sections of anti-synapsin-labeled midbrains revealed no gross anatomical differences in the neuropil structure or neuron bundles between *fruP1−* and wt males (Supplementary Movies 30 and 31).

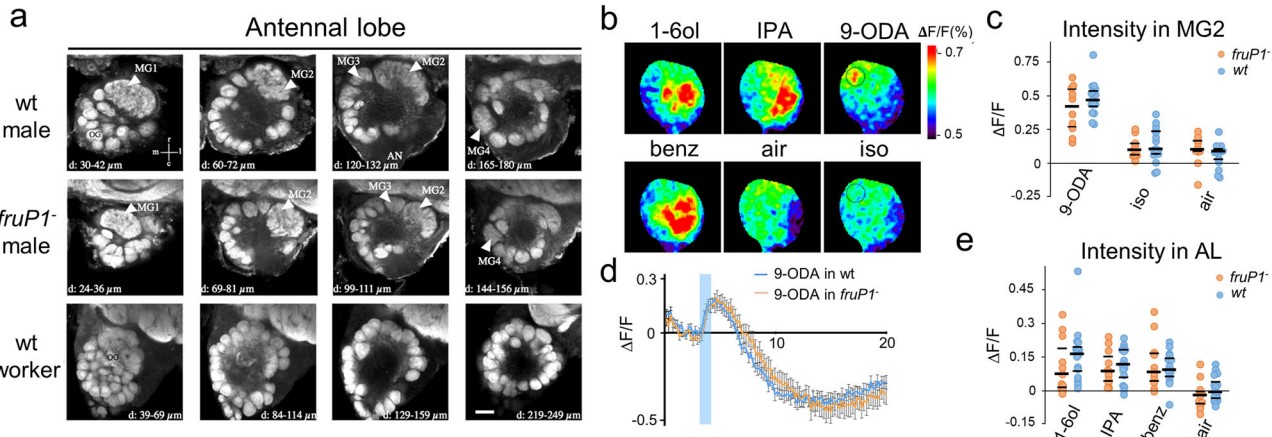

**Fig. 8 | Fru$^M$ protein activity is not required for sexual dimorphism or odor processing in the antennal lobe. a** Morphology of anti-synapsin-stained ALs. μm indicates the depths represented in the images. $n = 9$ *fruP1$^-$* and $n = 10$ wt males were examined. Arrowheads: macroglomeruli. Scale: 50 μm (**b**). Calcium signals in the antennal lobes of *fruP1$^-$* males evoked by 4 odorants (9-ODA, 1-hexanol (1-6ol), IPA and benzaldehyde). Different odorants induce different glomerular activity patterns. Relative fluorescence changes (Δ*F/F* [%]) are presented in a false-color code. Dark blue: minimal response. Red: maximal response. Dashed circle: macroglomerulus MG2. **c** Amplitude of calcium responses (Δ*F/F* [%]) in MG2 to sex pheromone 9-ODA and to air and solvent controls (iso: isopropanol). Medians (middle line) and quartiles are shown. No difference was detected between the *fruP1$^-$* (red) and wt (blue) groups (genotype effect: $P = 0.46$, $F_{1,24} = 0.57$, 2-way repeated-measures ANOVA. 9-ODA *fruP1$^-$* vs. 9-ODA wt males: $P = 0.36$, Sidak's multiple comparison test). In both cases, responses to 9-ODA were significantly greater than those to the controls (odor effect: $P < 0.0001$, $F_{2,48} = 65.25$, 2-way repeated-measures ANOVA. 9-ODA vs. iso control: $P < 0.0001$, Tukey post hoc tests. *fruP1$^-$* (red): $n = 10$. wt (blue): $n = 16$. **d** Average time courses (in seconds, x-axis) of odor-evoked responses (Δ*F/F* [%]) to 9-ODA recorded in MG2. *fruP1$^-$*: $n = 10$. wt males: $n = 16$. Blue bar: odor presentation. Calcium signals show a biphasic response, with increased fluorescence upon odor presentation. Mean values alongside with the Standard Error of the Mean (SEM) is presented. **e** Amplitude of calcium responses (Δ*F/F* [%]) to 3 odorants (1-6ol, IPA, and benzaldehyde) in the entire AL. Medians (middle line) and quartiles are shown. No differences in response were found between *fruP1$^-$* and wt males (genotype effect: $P = 0.49$, $F_{1,25} = 0.48$, 2-way repeated-measures ANOVA. Air: air control). Compared with the air control, all the odorants induced significant activity (odor effect: $P < 0.0001$, Tukey post hoc tests). *fruP1$^-$* (red): $n = 11$. wt (blue): $n = 16$. All statistical tests were two-sided.

Higher-resolution analysis of the AL, the first-order olfactory center, was performed. Four male-specific enlarged and structured glomeruli (MGs) were found in *fruP1$^-$* males that were in the same location as those in wt males (Fig. 8a)[14,18,58]. These results suggest that the gross brain anatomy and the sex-dimorphic structure of the AL were not malformed in *fruP1$^-$* males during development. Finally, we examined whether neural processing features were impaired in adulthood using in vivo Ca$^{2+}$ imaging of the AL. Odor information detected by OSNs in the antenna is translated into a glomerulus-specific pattern of neural activity in the AL[18,47]. For the queen pheromone 9-ODA, the activity in *fruP1$^-$* males was confined to MG2, as in wt males (Fig. 8b). The response intensity did not differ between *fruP1$^-$* and wt males ($P = 0.46$, 2-way repeated-measures ANOVA; 9-ODA *fruP1$^-$* vs. 9-ODA wt males: $P = 0.36$, Sidak multiple comparison test; Fig. 8c, d). The responses in MG2 were specific to the 9-ODA application in both cases ($P < 0.0001$, Tukey post hoc tests, 9-ODA vs. control, $P < 0.0001$; Fig. 8c, d). These observations suggest that 9-ODA odor information processing was not impaired in *fruP1$^-$* males in this first-order processing center. For other odorants, such as 1-hexanol (1-6ol), the repellent benzaldehyde, and the alarm pheromone component isopentyl acetate (IPA), we also detected no signal intensity or pattern differences between *fruP1$^-$* and wt males ($P = 0.49$, 2-way repeated-measures ANOVA; Fig. 8b, e), suggesting no dysfunction in *fruP1$^-$* males. Collectively, these results suggest that neither a dysregulated neuropil connectivity structure and chemoreceptor protein repertoire nor dysfunctional primary olfactory processing can explain, at this gross level, impaired behavior in *fruP1$^-$* males.

## Discussion
### The Fru$^M$ protein specifies initiation and sustainment processes for cooperative behaviors in the nervous system
The behaviors of male bees are well adapted to group living and colony benefits. A male does not behaviorally compete with other males over females or food stores in the protected shelter of the colony[8,9]. Instead, males receive nutrition-rich predigested food from worker bees during their maturation[10–12,50]. These trophallactic interactions among colony members thereby communicate the food status of the colony[10]. From these cooperative behavioral interactions, the colony benefits as a whole thereby increasing its reproductive success. Here, using loss-of-function mutation, we found that the initiation, choice, and sustainment aspects of male cooperative behaviors required *fruP1* transcripts, but not the stereotypic motor program elements of these behaviors. This *fruP1* transcript expresses a BTB zinc-finger transcription factor protein only in the male nervous system, suggesting that this Fru$^M$ protein determines the above aspects of innate behaviors. Specifically, we found that *fruP1* transcripts are required for the rate of approaching, begging, and trophallaxis behaviors, suggesting that the initiation of three sequences of the social feeding behavior is innately programmed by Fru$^M$. The duration of trophallaxis behavior also requires the Fru$^M$ protein, suggesting that there is a neural basis for a sustainment process that controls the duration of social feeding. The choice of whether males contact pollen stores in the cells also requires *fruP1* transcripts. This indicates that the Fru$^M$ protein programs repression of cell entering behaviors if pollen-derived stimuli are detected, which, as a consequence, reduces social conflicts over the collectively used food stores. Overall, we showed that Fru$^M$ specifies initiation and sustainment of program elements of cooperative behaviors in the nervous system allowing flexible control of social interactions.

In addition to its evolutionarily derived function, Fru$^M$ still has its ancestral function, the hardwiring of mating behaviors[59–62]. This conclusion is based on the impaired behavioral response of *fruP1$^-$* mutants to the queen pheromone 9-ODA, which attracts the males to the queen during mating flight[54,63].

### The initiation and sustainment processes innately scale the participation in the social feeding task
A key question is how the many colony members in the densely populated nest organize their social feeding task. Our results revealed

that the rates of approaching, begging, and the rate and duration of trophallaxis behaviors are innately programmed in males. These innate specifications have profound consequences for the colony as they innately scale the participation of individuals in the social feeding task. If males have a high rate and short duration of the social feeding behavior, worker bees will have small workload contributions to the collective feeding task. As a consequence, more interactions and more worker bees will be needed to accomplish the feeding of males. Conversely, if males have a low rate and long duration of the social feeding, each worker bee will provide a large workload contribution to the collective feeding and fewer worker bees will be needed to complete the collective task. Critically, this innate scaling of task participation for a given stimulus[64] departs from the classical view of task organization in which a bee directly respond and engages in a task behavior once the appropriate stimulus is encountered[65–68].

We suggest that an innate scale of task participation has important and general consequences for the organization of cooperative behaviors. (i) The scaling via rate and duration defines the average number of worker bees in the colony that engage and perform the task. (ii) The scaling via rate and duration also defines how the workforce of a single bee is split among different tasks. This is of importance, because a worker bee performs different tasks during a day[69,70]. (iii) The scaling via rate and duration also determines how many colony members are informed about the food status of the colony[10]. If a large fraction of colony members participates in social feeding, a large fraction is informed about the food status of the colony. As a consequence, a large fraction of colony members can adjust their task decisions to colony needs[69], when encountering task-related stimuli. Hence, we see that the innately scaled task participation links local stimuli encounter and individual task decisions with cooperative performances that establish benefits at the colony level.

## Fru^M+ circuitry processes, integrates and evaluates sensory information

Our understanding of how the cooperative aspects of behaviors are represented at the neural circuitry level is rudimentary at best[25–28]. The Fru^M protein is required for initiation and sustainment of social feeding behaviors and is expressed in a spatially restricted manner in the nervous system suggesting that the Fru^M-expressing circuitry mediates these aspects of behaviors. This Fru^M+ circuitry is involved in processing olfactory, gustatory and mechanosensory information, suggesting a possible role of different sensory modalities in the control of these within-colony behaviors[14,18,41,44,46,47]. Fru^M+ circuitries are also found in higher-order processing centers—the mushroom bodies—indicating that the circuitry is possibly involved in evaluating and integrating sensory information for behavioral decision processes[41,44,71–73]. Fru^M+ circuitry is also represented in elements processing visual information[42], which indicates possible involvement in the control of the mating flight behaviors. We also characterized elements of a sexually dimorphic Fru^M+ circuitries that possibly represent male-specific connectivity for the control of the male-specific behaviors. We found these male-specific neural populations in the antennal nerve and glomeruli, which are involved in odor processing, as well as in a neuronal tract involved in visual information processing[14,18,46].

How can the Fru^M proteins specify the male behaviors? We hypothesize that the Fru^M proteins determine the cell identity and the gross connectivity of Fru^M+ circuitries during development and/or the functional properties of these circuitries at adulthood. We found that this BTB zinc finger transcription factor does not regulate the repertoire of chemosensory receptor genes in the antennae[15,74], nor does it determine the gross connectivity structure of neuropils, nor does it specify the gross processing abilities in the AL[14,18,41]. The lack of mutant effect at these levels of organization suggests that we need a more spatially resolved analysis to characterize the programming

mechanism underlying behavioral specification that possibly also involves the higher-order processing centers.

## Programming of cooperative behaviors derived from sexual behaviors

It is largely unknown how cooperative behaviors evolve through a genetically encoded program. Here, we demonstrate that coopting the *fru* gene from its ancestral, reproductive behavior-specific function[59–62] represents an evolutionary mechanism involved in the origin of cooperative behaviors. These cooperative behaviors inside the nest evolved with the origin of colony formation and eusociality in the honeybee lineage[19,20]. Previous studies revealed the molecular rate, genome evolution, gene family expansion, signatures of selection, and changes in the gene regulatory network at the genome scale[21–24] that are associated with the origins of sociality and eusociality. In this study, we identified a single gene by functional perturbation that programs initiation and sustainment processes of within-colony cooperative behaviors. This programming now allows dissection of evolutionary mechanisms underlying social organization at the level of the nervous system. The *fru* gene also plays a role in hardwiring mating behaviors in honeybees, as in other insects. They all use the same mechanism of sex-specific *fruP1* transcript splice regulation, which suggests a deep conservation of *fru's* role in specifying sexual behaviors[59–62]. Most remarkably, the number of Fru^M-expressing cells in the midbrain is approximately conserved between honeybees (in this study: 1800) and the solitary-living fruit fly *D. melanogaster* (1700)[75] despite a larger brain in honeybees[16]. In *D. melanogaster*, Fru^M-expressing OSNs, which detect female pheromones, project into three glomeruli that are specifically enlarged in males[76–78]. The formation of these sexual dimorphic glomeruli depends on the male-specific activity of Fru protein[76–78]. To the contrary, Fru^M in honeybee males is not expressed in the OSNs that enter the glomeruli (Fig. 4a, b), is not required for the sexual dimorphic development of the male-specific enlarged macroglomeruli (Fig. 8a), and the processing of queen's pheromone information at the level of macroglomerulus MG 2 (Fig. 8b–d). These differences revealed that Fru^M's role diverged in the two species, suggesting notable entry points to dissect honeybee-confined specifications of the circuitry and cooperative behaviors. Collectively, these findings exclude that the increase in the number of Fru^M+ neurons is an evolutionary mechanism underlying the origin of cooperative in honeybee males. Rather, the comparisons suggest that the connectivity and functional properties of Fru^M+ circuitry diverged in the species, possibly involving the evolution of other downstream controls of Fru protein target genes. Hence, the Fru^M+ circuitry in the honeybee offers a great opportunity to understand the control and evolution of cooperative behaviors at the level of the nervous system.

## Methods

### Honeybee source

Honeybee colonies (*Apis mellifera*) were located in the bee yard or in the containment at the Heinrich-Heine University Düsseldorf, Germany. The genetically manipulated honeybees were maintained together with worker bees in small mating nucleus hives (Segeberger nucs), which we kept in a secure containment so that genetically manipulated animals could not escape into nature.

### RNAi mediated knockdown procedure

Female eggs were injected with fem siRNAs, and larvae were reared to 4th instar (L4) inducing a temporally restricted knockdown as described in ref. 31.

### CRISPR/Cas9 experimental procedures

Target sites for the sgRNAs were identified using Benchling software (Biology Software, 2017, https://benchling.com). The sgRNAs for the

deletion were fruP1 sgRNA 6 (GATACCCCCCACGACATTTC) 855 bp upstream of the designated P1 transcript start, and sgRNA 12 (GGCTGCTGTGCACGCTTAGA) 232 bp downstream of the translational start codon (Fig. 1C). The deletion rate was 14%. The sgRNA for the genomic insertion was fru sgRNA 1 (GTGTTTGGCGCATCGTTACC) 22 bp downstream of the start codon. sgRNAs were synthesized[29]. Female eggs were collected from wild type queens every 90 min and injected directly with 53-mm injection needles (Hilgenberg, Malsfeld, Germany)[29,79]. We injected 375 ng/µl of Cas9 protein (EnGen Cas9 NLS, S. pyogenes, #M0646, New England Biolabs), sgRNA1 and donor DNA were at a molar ratio of 1:1:1. The myrGFP-P2A-T2A coding sequence was synthesized with fru homologous arms (Supplementary Fig. 5; Eurofins, Ebersberg, Germany). The homologous arms (each 250 bp long) corresponding to the intron 1/2 and exon 2 [National Center for Biotechnology Information (NCBI); gene ID: 409022; reference Sequence: NC_037646; Assembly: Amel_HAv3.1 (GCF_003254395.2)]. In the inserted sequence, we modified the PAM site of the fru sgRNA 1 target site[39]. Queens were reared, and genotypes were determined via PCR amplification, size determination, and sequencing[39,64,79]. Mutant queens were treated twice with $CO_2$ (on day 9 and day 10) to stimulate oviposition of unfertilized, male eggs. Queens were introduced into small colonies and nucs (Holtermann, Germany).

## Immunohistology and image acquisition procedure

We developed for the honeybee an antibody against Fru protein involving part of the BTB domain. The protein for immunization (Seqlab, Göttingen, Germany) in rabbits was 98 amino acid long (VSQH LLPMFLKTAEALQIRGLTDNSVNNKTEEKSPSPEPETQTGIRHTESPNLQPP PEKRKRKASGSYDVSLSGPPSERFMSDSQTSSQCSYKSSPPV) and is present in all protein isoforms (Fig. 1b–d). Anti-Fru antibodies were purified from the serum using HiTrap NHS-activated HP columns (GE Healthcare Life Sciences, Freiburg, Germany). For whole mounts, brains were dissected in ice-cold honeybee saline (130 mM NaCl, 5 mM KCl, 4 mM MgCl2, 5 mM CaCl2, 15 mM Hepes, 25 mM glucose, 150 mM sucrose, pH 7.2) and fixed in 4% ice-cold formaldehyde (Roth, Karlsruhe, Germany) in phosphate-buffered saline (PBS, pH 7.2) at 4 °C for at least 24 hours. The brains were washed 3 × 10 min in PBS, 1 × 10 min in PBS containing 2% Triton X-100 (PBS-T) and 2 × 10 min in 0.2% PBS-T. All washing steps were performed at room temperature on a shaker. Brains were blocked at room temperature for 1 h in 0.2% PBS-T containing 2% normal goat serum (NGS), and incubated with 0.2 units of Alexa Fluor 568 phalloidin (Molecular Probes, A-12380, Eugene, USA) and rabbit-anti-Fruitless antibody (1:800 to 1:2000) in blocking solution for 4 days at 4 °C. In case of fru^myrGFP male brains, chicken-anti-GFP antibody (1:1000; Rockland Immunochemicals, Inc., Limerick, PA, USA) was added. Next the brains were washed 3 × 1 h and incubated with goat-anti-rabbit secondary antibody (1:250 to 1:800; Fisher Scientific, Schwerte, Germany) in 0.2 % PBS-T with 2 % NGS for 2 days at 4 °C on a shaker. In case of fru^myrGFP male brains, goat-anti-chicken antibody (1:250; Fisher Scientific, Schwerte, Germany) was added. After washing 4 × 5 min in PBS, the brains were dehydrated in an isopropanol series (10, 30, 50, 70, 90% isopropanol in PBS and 2 × 100 % isopropanol, 5 min each step) and subsequently cleared in methylsalicylate (MS; 99%; Sigma Aldrich, Steinheim, Germany) before mounting in fresh MS and storing at 4 °C in the dark until imaging.

For cryosections, the dissected brains were fixed in 4% ice-cold formaldehyde (Roth, Karlsruhe, Germany) in phosphate-buffered saline (PBS, pH 7.2) with 8% sucrose at room temperature (RT) for 90 min. Fixed brains were washed 3 × 10 min in PBS at RT, followed by 10% (30 min) and 20% (30 min), and 30% sucrose buffered PBS solutions (overnight). Brains were frozen in tissue imbedding media on dry ice. Cryosections (10 µm and 20 µm) were washed 3 × 10 min at RT in PBS-T followed by overnight anti-Fru antibody staining (1:10000) at 4 °C using PBS-T and 0.1 % bovine serum albumin (BSA). After washing 3 × 10 minu at RT in 0.1 % PBS-T, the cryosections were incubated with the secondary antibody (donkey-anti-rabbit Cy3 (1:200) in 0.1 % BSA, 0.1% PBS-T for 90 min and replaced 0.1 % BSA, 0.1 % PBS-T with dye Hoechst34580 (100 ng/ml) for 30 min at RT. Cryosections were washed 3 × 10 min and embedded into glycerol-propyl-gallate. For detailed anatomical information, immunostaining of antennal lobes was performed after calcium imaging experiment (Fig. 8a). Dissected brains were immediately immersed in cold 1% zinc formaldehyde in PBS[80] and kept overnight at 4 °C. Brains were then washed six times in PBS (10 min each), permeabilized using 1% PBS-T for 30 min, and preincubated for 3 h in 1% BSA, 0.3% PBS-T. The brains were then incubated in 0.1% BSA 0.3% PBS-T, and mouse monoclonal anti-SYNORF1 (DHSB, #3C11, US) at 1:100 dilution for 7 days. Brains were then washed 6 times in 0.3% PBS-T and secondary anti-mouse antibody (1:200) coupled to Alexa 555 (Thermofisher, #A-21147, France) for 5 days. Brains were then washed in PBS, dehydrated in an ascending ethanol series (30% to 100%), cleared and finally mounted in MS.

Confocal images of wholemounts were captured on a Leica TSC SP8 STED 3X (Leica Microsystems, Wetzlar, Germany). Optical sections for all z-stacks were taken at  µm intervals with a resolution of 512 × 512 pixels. We generated image z-stacks of one hemisphere of midbrains and ganglia. In addition, we generated tile scans of whole brains, which were merged using the processing tool Mosaic Merge of LAS X (Leica Application Suite X 3.0.0, Leica Microsystems CMS, Wetzlar, Germany) with a 20x objective (multi/ NA 0.75). Additionally, z-stacks of selected brain regions were imaged using a 40x objective (water/ NA 1.10).

Cryosections were imaged with a laser-scanning confocal microscope (Zeiss LSM510, Carl Zeiss Microscopy, Jena, Germany). For antennal lobe anatomical details after imaging, lobes were scanned using a laser-scanning confocal microscope (Zeiss LSM 700) with a W Plan-Apochromat 20x/NA 1.0. Images were processed using FIJI (ImageJ 1.53c; Wayne Rasband, National Institutes of Health, USA).

## DNA and RNA preparation and PCR procedures

Genomic DNA was isolated using innuPREP DNA Mini Kit (Analytik Jena, Jena, Germany). Induced mutations were characterized by PCR amplification using high fidelity polymerase (Phusion High-Fidelity DNA Polymerase (Thermo Scientific)[81]. Amplicons of different progeny (haploid males) were cloned and double strand Sanger sequenced (Eurofins, Ebersberg, Germany) using at least three different clones per sample. RNA was isolated using a Trizol-based protocol[33]. First-strand cDNA was synthesized using the RevertAid First Strand cDNA Synthesis Kit (Thermo Scientific). P1 transcript was amplified using oligonucleotide primers SK60/SK61 (SK60: AACTGATCCTCCTTCCGTGCTG CG, SK61: AGTGGTTCCTGATGTGCGTCACGA) for female P1 transcripts and SK73 & SK61(SK73: TCGCGATGCTACGTCAACTGTAGG) for male fru P1 transcripts (Eurofins Genomics, Ebersberg, Germany). Semiquantitative RT-PCR amplifications were run under nonsaturating conditions and in technical triplicates for each sample using elongation factor 1α (ef1α) gene for semiquantitative adjustment across samples (GATATCGCCCTGTGGAAGTTC (for 632 bp) or GTGAT CGGCCACGTTGATTC (for 800 bp) together with GCTGCTGGAGCG AATGTTAC)[32]. Amplicons were resolved by agarose gel electrophoresis and stained with ethidium bromide. The uncropped gel pictures are presented in the Source Data file or as a Supplementary Fig. The 5′ and the 3′ of the various fru transcripts were identified by RACE experiments using mRNAs from 2 to 4-day old male and female pupal heads[31]. Other intron/exon structure was determined by mapping the sequences derived from various RT-PCRs to the genomic sequence [National Center for Biotechnology Information (NCBI) Assembly: Amel_HAv3.1 (GCF_003254395.2)]. Sequence data of the transcripts are available under EMBL-EBI accession number: OZ362630-OZ362641. For the RNAseq we generated three pools of wt and three pools of fruP1⁻ male antennae (each pool with antennae from 5 individuals).

RNA was initially isolated using Trizol, and further purified using RNeasy MinElute Cleanup Kit (Qiagen, Hilden, Germany)[33]. Total RNA samples were quantified (Qubit RNA HS Assay), and quality was accessed using the Fragment Analyzer and the "Total RNA Standard Sensitivity Assay" (Agilent Technologies). Library preparation was performed according to the manufacturer's protocol using the "VAHTS Universal RNASeq Library Prep Kit for Illumina® V6 with mRNA capture module version 7.0" (Vazyme Biotech Co.). 500 ng total RNA was used for mRNA capturing, fragmentation, the synthesis of cDNA, adapter ligation, and library amplification. Bead-purified libraries were normalized and finally sequenced on the HiSeq 3000/4000 system (Illumina Inc.) with a read setup of SR 1 × 150 bp. The bcl2fastq2 tool (v2.20.0.422) was used to convert the bcl files to fastq files as well for adapter trimming and demultiplexing. 4–12 million single-end reads with a length of 150 bp (EMBL-EBI accession number: PRJEB100586) were mapped to the *Apis mellifera* transcriptome (NCBI Assembly Amel_HAv3.1_rna.fna) using the kallisto software tool[82]. Estimated read counts were normalized using the transcripts per kilobase million (TPM) method[83]. Differences in gene expression were calculated using DESeq2[84]. Genes were differentially expressed (DEGs) if adjusted P-values (Padj) were <0.05 (Wald test) and log2 fold change was greater than 1.5.

## Behavioral analysis in colonies and petri dishes

Capped male brood from *fruP1⁻/⁻* and wt queen colonies were placed in separate nucs in an incubator at 34 °C along with capped wild-type worker brood frames and a comb containing honey and pollen. *fruP1⁻* and wt adult males were color-coded to keep track of the age. The adult workers were removed from the nuclei each morning.

For the within-colony experiments, newly emerged (0–24 h old) *fruP1⁻* males, wt males, and worker bees were each tagged with unique 2D barcodes. 20 *fruP1⁻*, 20 wt males, 460 worker bees, and a queen were transferred to comb containing the same amount of honey and pollen in same location in each replicate[64] (Supplementary Fig. S7). Each of the two pollen store areas contained 15 g of pollen ("Echter Deutscher Spezial Blütenpollen", Werner-Seip-Biozentrum GmbH & Co. KG, Butzbach, Germany), which we added to the cells together with 25 µl of sugar syrup ("Ambrosia Futtersirup", Nordzucker AG, Braunschweig, Germany). The two honey store areas consisted of 225 cells, each cell filled with 200 µl of sugar syrup. This small colony was kept overnight at 34 °C, and tracking was run lasted for 24 h at RT. We performed two tracking experiments (August 25th and September 8th, 2021) on males that were 2–5 days old. Tags were computer-based tracked using the Bee Behavioral Annotation System (BBAS[49]). Bee trajectories were not continuous, and gaps resulting from single missing frames, were linearly interpolated[64]. To measure duration (minutes spent in area/hour) and number of visits (visits/hour) in the different work areas and the walking distance (meters/hour), we used our C++ and Java programs scripts[64]. We thereby included only hours in which the detection rate was more than 10%. Only bees that had more than 12 of those hours per day were included in the analysis. To follow the behavioral repertoire of male bees in the colony, the tracking data were overlaid on the video using VirtualDub software (VirtualDub-1.9.11, virtualdub.org)[64]. This allowed us to continuously track individually tagged males throughout the observation period, even when the barcode was not automatically recognized. Behavior of *fruP1⁻* and wt males was examined for 15-min periods between noon and 3.30 p.m. The examinations were randomized and blinded in respect to the genotype. Individuals who rested for more than 40% of the time during the observation period (possibly a sleeping period) were not included in the analysis. Male behaviors tend to follow stereotypical patterns and were categorized according to a previous study[8]. If a male did not

move for at least 5 s, this was defined as resting behavior. If a male was disturbed by other bees and moved to a new spot, but rested again within 10 s, we counted this as same resting behavior event. Self-cleaning behaviors include (i) head cleaning (the male uses its forelegs to brush the head), and (ii) body cleaning (the male uses its hindlegs to brush its abdomen). We also observed that males entered with their heads the cells on the comb. In some cases, this occurred repeatedly in short bouts of cell entering activities. Duration of entering a cell was defined as the time the head was in the cell. The males were actively interacting with other colony members. The male contacted other colony members either using their antennae and/or proboscis. In the case of begging behavior a male contacted a worker's head with either its antennae or its proboscis. If the male contacted the abdomen or the thorax of either a worker or a male, we defined this as other social contact behavior. During trophallaxis behavior male's glossa protruded between the worker's mandibles to obtain the liquid food. Thereby, the male's forelegs often tapped the worker and the male moved its head forward.

General sensorimotor functions of males were examined in petri dishes as previously reported[64]. The following changes were applied. A rectangular petri dish (12 cm²) with 5 mm (diameter) ventilation holes was used. Males were introduced into the arena 5 min prior to the start of the test. Males were older than 2 days. Locomotion responses to odorants and worker bees were quantified by calculating the differences in line crossings between control condition (2 min) and the treatment (2 min) for each male. To quantify the odor response, we introduced an odorant into the arena via a filter paper (75 mm, grade 413; VWR, International GmbH, Darmstadt, Germany) through a ventilation hole. For the 9-oxo-2-decenoic acid (9-ODA) response, the locomotion behavior with respect to control filter paper (1 µl isopropanol) and 9-ODA filter paper (1 µl of 9-ODA; 50 µg/µl, dissolved in isopropanol) was examined. Filter papers were only introduced into the arena after the isopropanol had evaporated for 1 minute. The benzaldehyde response has been previously described[64]. To quantify the male's locomotion response to social contacts, we introduced into the arena two worker bees to the male and compared line crossings 2 min before and 2 min after the introduction. Petri dishes and papers were changed after each bee was tested. We used a camera (60fps, Full HD, 44100 Hz; Casio Exilim Pro EX-F1) to count line crossings in the video recordings obtained using VSDC Free Video Editor (Multilab LLC). The proboscis extension response (PER) was examined using honey that was presented to antennae of immobilized males[85]. Whether a male responded or not was examined every two 2 hours for a maximum of four times. All examinations were randomized and blinded in respect to the genotype.

## Anatomical analysis procedures

Images of the gross anatomy were taken using a binocular (SteREO Discovery.V12, Zeiss) with an attached camera (Axiocam 208 color, Zeiss) and 3D software (ZEN blue, Zeiss). Reproductive organs (from 1 to 2 days old males) or midgut and hindgut (from 2-day old males kept on a pollen/honey comb) were dissected and imaged using a binocular (S8 APO, Leica) with an attached camera (UI-1240LE-C-HQ, IDS Imaging Development Systems) and the uEye Cockpit software (IDS). Content of the gut was characterized by dispersing the midgut and hindgut in 80 µl H₂O. The number of pollen grains in 40 µl of this suspension was counted under a microscope (Axiovert 25 CFL, Zeiss) using microscope slides.

## Cuticular hydrocarbon analysis procedures

Surface lipids were extracted from individual bees by submerging bodies in 0.5 ml of n-hexane for 1 min. Washes were analyzed by GC/MS using a HP5890II gas chromatograph coupled to a HP5972 mass spectrometer (Hewlett Packard, Palo Alto, California, USA), equipped

with a DB-5MS column (30 m, 0.25 μm film thickness, 0.25 mm ID), with splitless injection. The GC oven was programmed from 60 to 300 °C at 10 °C/ min followed by 15 min at 300 °C. Peaks were called using a standardized integration threshold in ChemStation Software (Agilent Technologies, Santa Clara, California, USA). Integrated ion currents (peak areas) of all linear and methyl-branched hydrocarbons (CHC) were used for downstream analysis.

## in vivo calcium imaging procedures

Eclosed mutant males were maintained in hives for a minimum of 8 days and a maximum of 15 days. On the day of the experiment, the males were chilled on ice for 5 min just prior to the experiment until they stopped moving. Then, they were prepared following the standard preparation used to image the ventral part of the honey bee brain[86]. Glands as well as trachea were removed to expose the brain, and the pool was filled with ringer solution (in mM: NaCl, 130; KCl, 6; $MgCl_2$, 4; $CaCl_2$, 5; sucrose, 160; glucose, 25; Hepes, 10; pH 6.7, 500 mOsmol; all chemicals from Sigma-Aldrich, France), to avoid desiccation of the brain surface. For staining, the saline solution was gently removed, and the brain was bathed with 20 μL of dye solution (10 μg Oregon Green 488 BAPTA-2 AM dissolved with 4 μl Pluronic F-127, 20% in dimethyl sulfoxide, all from Molecular Probes, Invitrogen). The bee was left for 45 min in a humid and dark place, and then the brain was rinsed again thoroughly with saline solution in order to remove extracellular dye.

A T.I.L.L. Photonics imaging system (Martinsried, Germany) was used to perform in vivo optical recordings, as described elsewhere[87,88]. An epifluorescence microscope (Olympus BX51WI) was used to record activity in the AL using a 10× water-immersion objective (Olympus, UMPlanFL; NA 0.3). Oregon Green was excited using 488 nm monochromatic light (T.I.L.L. Polychrom IV). Fluorescence light was separated by a 505 nm dichroic filter and a long-pass 515 nm emission filter, and recorded with a 640 × 480 pixels 12-bit monochrome CCD camera (T.I.L.L. Imago) cooled to −12 °C with 4 × 4 binning on chip. Each measurement consisted of 100 frames recorded at a rate of 5 Hz (integration time for each frame ~50 ms). To test the odorant stimuli, a constant airstream was directed from a distance of 1 cm to the male's antennae, and stimuli were given at the 15th frame for 1 second (5 frames correspond to 1 second). For each odorant stimulus (all obtained from Sigma-Aldrich, France), 5 μL of the solution was deposited on a filter paper inserted in a Pasteur pipette. We tested a small set of 3 volatile odorant stimuli known from previous work to trigger strong neural activity in workers: 1-hexanol (1-6ol), isopentyl acetate (IPA), and benzaldehyde. We also recorded responses to 9-ODA at a concentration of 50 μg/μL. As control stimulus, a pipette containing the solvent, isopropanol (iso), or a clean piece of filter paper was used. Each odorant stimulus was presented twice in a pseudo-randomized order. Imaging data were processed and analysed using custom-made software written in IDL 6.0 (Research Systems, Boulder, CO). First, the relative fluorescence changes were calculated as $\Delta F/F = (F − F0)/F0$ by taking as reference background F0 the average of three frames just before the odorant stimulation (frames 9–11). Possible irregularities of lamp illumination and bleaching were corrected by subtracting the median pixel value of each frame from every single pixel of the corresponding frame. Finally, the two spatial dimensions were filtered with a Gaussian filter of window size 7 × 7 pixels for noise reduction. A biphasic calcium signal was observed in all recordings. For the quantification of response intensity, a mask was precisely drawn in order to exclude regions outside of the imaged structure, and the response intensity was calculated by averaging the intensity of all pixels located within the unmasked area and the two presentations of each odorant. The intensity of the odor-induced response was obtained by averaging three consecutive frames at the end of the odor presentation (frames 19–21) and subtracting the average of 3 frames during the negative

component of the signal (frames 49-51). All results are displayed as means over individuals ± SEM.

## Statistical analyses procedures

Mann–Whitney $U$ test, one-sample Wilcoxon and log-rank tests were performed using the SPSS 29 and Systat 13 software. Testing was performed two-sided. The compositional similarity of CHC profiles of individual bees was visualized by nonmetric multidimensional scaling (n-MDS) and analyzed using one-way analysis of similarities (ANOSIM, with pairwise tests) using Primer v.6 software[89]. ANOSIM is a non-parametric permutation statistic that tests whether a factor affects the rank order of pairwise similarities in a similarity matrix (here Bray–Curtis similarities of CHC composition between individual bees). The resulting $R$ value ($−1 < R < 1$) indicates the degree of separation between treatments (complete separation: $R = 1$, no separation: $R = 0$). To identify compounds that contributed most strongly to overall dissimilarity between treatment groups, we used the SIMPER module, also in Primer v.6. n-MDS. ANOSIM and PRIMER analyses were based on standardized (percentage), square root transformed GC/MS integrated ion currents (peak areas). For calcium imaging recordings, odor response intensities were compared with ANOVA for repeated measurements, using odorants as within-group factors. A Dunnett post hoc test was applied to compare the intensity of the response to each stimulus with a common reference, the air control. All tests were performed with GraphPadPrism V7.00 and R (version 4.4.0, www.r-project.org).

## Reporting summary

Further information on research design is available in the Nature Portfolio Reporting Summary linked to this article.

## Data availability

Nucleotide sequence data are openly available from the European Nucleotide Archive (ENA) at EMBL-EBI under accession number PRJEB100586 (RNAseq data) https://www.ebi.ac.uk/ena/browser/view/PRJEB100586 and accession number OZ362630-OZ362641 (*fru* gene cDNA data) https://www.ebi.ac.uk/ena/browser/view/OZ362630-OZ362641. The uncropped gel pictures are presented in the Source Data file. The authors declare that other data supporting the findings of this study are available within the paper and its supplementary information files. This study did not generate new or unique reagents. Source data are provided with this paper.

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

## Acknowledgements

We thank M. Müller-Borg for assisting with bee handling and molecular analysis support. We thank M. Müller for the development of the antibody and the experimental work. We thank A. Schönberg for providing technical support and M. Griese for providing bee colonies. We thank R. Raub for assistance with the high-performance computer cluster and the "Center for Information and Media Technology" (ZIM) at Heinrich Heine University (HHU) for computational support. We thank the many Master and Bachelor students for assisting with data collection. We thank S. Goodwin for further inputs on the manuscripts. Amplicon sequencing and sequencing quality control were performed by the Biological and Medical Research Center (BMFZ) at the HHU. We thank the Center for Advanced Imaging (CAi) at the HHU for providing the confocal microscope. JC and JCS thank the University Paris Saclay and the CNRS.

## Author contributions

S.K. and M.B. conceived the project. S.K. and M.B. designed the experiments. S.K. performed most of the genetic and behavioral experiments and brain scans, analyzed the data, and developed the figures. P.U. performed single-bee behavioral analysis, determined genotypes, scanned brains, and supported the data analysis and the design of the figures. A.S. analyzed the GFP-labeled brain scans and developed one of the figures. J.C. and J.-C.S. performed the Ca imaging experiments and scans of the antennal lobes. T.E. performed the CHC examinations. S.K. and M.B. wrote the manuscript. All the authors discussed the data, the results, and the manuscript. The project was funded by the Deutsche Forschungsgemeinschaft (http://dfg.de/): BE 2194 to MB.

## Funding

## Competing interests

The authors declare no competing interests.
