## [Transparent Peer Review file · Nature Communications]

The fru gene specifies male cooperative behaviors in honeybee colonies

Corresponding Author: Professor Martin Beye

Version 0:

Reviewer comments:

Reviewer #1

(Remarks to the Author)

The manuscript by Köhnen et al presents molecular, morphological, and behavioural evidence for the involvement of the fru gene in shaping the feeding behaviour of male honey bees. This study makes a valuable contribution to our understanding of the molecular underpinnings of social behaviour. The ms is well written, clearly presenting the results of elegant experiments and concluding with balanced, insightful discussion. I have only a few minor suggestions, as outlined below:

Introduction

Lines 71-72: This paragraph should not begin with “however,” because this word typically contrasts with a statement just made, and here the contrast is with a point from the previous paragraph (again in line 102). More importantly, the ms emphasizes the lack of information regarding the molecular mechanisms controlling cooperative behaviours in honey bee males. The behaviours described so far primarily focus on the male-worker interaction where males receive food -a one-way interaction that provides an immediate benefit only to the males. Please also highlight the reciprocal aspect by stressing the indirect benefits that workers (and the colony as a whole) gain from trophallaxis, thereby presenting a clearer view of the studied “cooperative” behaviour.

Line 77: The phrase “Compared with worker bees” is unclear. Could the authors clarify whether males possess the pathway while workers do not? Please provide more details.

Results

Line 115 (and figure 1): Are P1-P3 designated as promoters or as transcriptional start sites? Please clarify their functional roles based on the specific nucleotide sequences or motifs used to define or identify these elements. Additionally, Figure 1c is lacking the female protein representation, which is expected to be strikingly different to that of males. In lines 122-123 include the information on the expected peptide size of the female counterpart.

Figure 1e: Are the levels of ef1 α in wt males exactly the same to those in females?

Lines 431-434: I recommend softening this statement, as the type and extent of the morphological analyses conducted do not provide sufficient support for such a strong conclusion.

Discussion

Lines 532-536: This idea has been reiterated throughout the manuscript -initially in the Introduction, followed by the Results, and now in the Discussion. I recommend elaborating on it more thoroughly in the Introduction, particularly emphasizing its benefits for the colony as a whole, as previously suggested.

Lines 616-619: Has the fru gene been specifically identified as associated with social evolution in any of the 4 papers cited by the authors? If so, do these studies propose any common or divergent mechanisms of action for the gene?

Overall, the ms would benefit from punctuation and spelling editing (e.g. lines 87, 327).

Reviewer #2

(Remarks to the Author)

The manuscript titled “The fru gene specifies male cooperative behaviors in honeybee colonies” presents compelling and novel findings that significantly advance our understanding of the genetic control underlying social behaviors in honeybee colonies. This research provides insightful data that builds upon existing knowledge of social behavior in insects, with a particular focus on the role of genetics in shaping these behaviors. By exploring the fruitless (*fru*) ortholog in *Apis mellifera*, the authors uncover essential facets of male development and interaction within the hive.

Their investigation reveals that the *fru* gene expresses male-specific products during the late developmental stages of the nervous system, highlighting a critical window in which genetic influences can shape behavioral outcomes. This expression is particularly noteworthy because it suggests that the *fru* gene is active at key points in male maturation, potentially influencing the development of behaviors necessary for reproductive success and social integration within the colony. These findings align with observations in *Drosophila* and other dipterans, where the *fru* gene has also been implicated in regulating mating behaviors, demonstrating a conserved mechanism across different species.

However, the manuscript goes beyond mere correlation and establishes a direct link between the male-specific *fru* products and various behavioral paradigms within honeybee colonies. Notably, these male-enriched *fru* products are shown to be essential not only for modulating male courting behaviors but also for regulating social feeding behaviors that are vital for the colony's overall function. This dual role of the *fru* gene suggests a complex interplay between mating and cooperative social interactions, providing a deeper understanding of how evolutionary pressures may have shaped these behaviors to promote colony survival.

The research illustrates how the expression of male-specific *fru* products is crucial for drones to exhibit typical behaviors such as normal rates of begging for food, the duration of trophallaxis—an important process in which food is exchanged among colony members—and the dynamics of interactions with worker bees. These interactions are critical, as they help to maintain the nutritional health of the colony and ensure a cohesive social structure. The results underscore the fundamental importance of these behaviors in the daily functioning of the hive, emphasizing the role of males beyond reproduction.

A particularly striking aspect of the study is the documentation of the consequences of manipulating *fru* gene expression.

The authors provide compelling evidence that drones with a specific disruption in the expression of the *fruM* isoform exhibit a dramatic two- to threefold decrease in the execution of essential social behaviors, such as begging and trophallaxis. This finding provides robust support for the hypothesis that the *fru* gene is a key regulator of male-specific behaviors that facilitate not only individual reproductive opportunities but also the overall health and efficiency of the colony.

In summary, this manuscript significantly enriches our understanding of the genetic underpinnings of social behavior in honeybees, illuminating the intricate connections between genetics, behavior, and social structure within colonies. The implications of this research extend beyond honeybees, potentially informing studies on behavioral genetics in other social insects and adding depth to our comprehension of the evolutionary biology of complex social systems. By elucidating the multifaceted role of the *fru* gene in male-specific behaviors, this work sets the stage for future investigations into the genetic and molecular mechanisms that orchestrate social interactions in eusocial species.

To further improve the manuscript, we propose minor revisions and request the authors to address the following points:

Previous studies in *Drosophila* have demonstrated that male expression of the fruitless gene is not only necessary for male courtship behaviors but is also sufficient to induce male-like behaviors in otherwise normal females (Demir & Dickson, 2005, Cell). This significant finding led to the conclusion that the *fru* gene plays a central role in specifying various aspects of complex innate behaviors. Given this context, we are interested in understanding why the authors did not incorporate gain-of-function experiments—specifically, the expression of *fruM* in the brains of female workers through methods such as deleting the female-specific exon—to rigorously test whether *fruM* is sufficient to activate male-like cooperative behaviors in otherwise normal female workers.

This omission is particularly intriguing as it presents an opportunity to explore the sufficiency of *fruM* in reshaping behavioral phenotypes traditionally associated with male *Drosophila*. By examining the potential to induce male-like behaviors in females, the results of such experiments could significantly enhance our understanding of how *fruM* integrates with existing neural circuits involved in cooperative behavior. Furthermore, additional insights could clarify the genetic and neurobiological underpinnings of sex-specific behaviors, potentially revealing broader implications for understanding the evolution of behavioral traits across species.

General Comments and Suggestions for Revision:

1. Title Revision: We propose changing the title from “The fru gene specifies male cooperative behaviors in honeybee colonies” to “The fru gene contributes to male cooperative behaviors in honeybee colonies” to better reflect the data, which demonstrate necessity but not sufficiency.
2. Missing Gain-of-Function Evidence: The authors provide strong loss-of-function data using CRISPR/Cas9. However, gain-of-function experiments (e.g., ectopic expression of *fruM* in females by deleting the female-specific exon) would be crucial to test sufficiency. We request that the authors comment on why this was not attempted.
3. Reference to Previous Work:
 - The authors should cite and compare their findings on *fru* gene structure and splicing isoforms in *Apis mellifera* with *Nasonia vitripennis* (Bertossa et al., 2009, Mol Biol Evol. PMID: 19349644).
 - Additionally, they should reference Siwicki & Kravitz (2009, Curr Opin Neurobiol) for background on *fru* and *dsx* in *Drosophila* behavior.
 - Consider also Gailey et al. (2006, PNAS, PMID: 16319090) regarding the evolutionary conservation of fruitless function.
4. Clarification on Expression and Antibody Specificity: The manuscript notes that the antibody detects both male-specific *FruM* and a common isoform. The authors should clarify if the common isoform is undetectable or weakly expressed in both sexes, and whether P2/P3 promoters are active in the brain.
5. Discussion of Additional Genetic Contributors: While the paper attributes behavioral phenotypes to *fruM* loss, the observed partial phenotypes suggest other genes likely act in parallel. The authors should discuss candidate genes or

regulatory pathways that might also contribute.

6. Potential Role of *dsx*: *Drosophila* studies show that *dsxM* and *fruM* act in concert to establish male courtship behavior. The authors should comment on why they did not knock down *dsxM* alone or in combination with *fruM*.

7. Clarification of RNAi and Developmental Timing:

- The authors should explain why *fruM* mRNA was detected in fem RNAi-treated female larvae alongside substantial female-specific *fru* transcripts.
- They should also provide the developmental timeline (egg to L4 larva), duration of RNAi effect, and siRNA sequences used.

8. RT-PCR Band Size Discrepancy: In Fig. 1 panels e and f, *fruM* and *ef1a* bands differ in size. The authors should clarify this, providing primer sequences and amplicon locations.

9. Clarify Methodological Details:

- sgRNA design and efficiency for CRISPR deletions should be reported.
- Clarify ambiguous phrases such as “Numbers: larval and pupal stages,” and rephrase statements like “it is unclear where *FruM* is expressed” (when prior data showed expression).

10. Gene-Centric Language: Several statements over-attribute behavioral control directly to a single gene. For example:

- “*FruM* specifies decision processes...” should be changed to “*FruM* facilitates decision processes...”
- “...programs decision processes...” should be softened to “contributes to the neural basis of...”
- These revisions would avoid an overly reductionist interpretation and better reflect the multifactorial nature of behavior.

11. Abstract and Introduction Revisions:

- Line 35: Replace “whether” with “how” — “but how cooperative behaviors require...”
- Lines 39–40: The claim about screening transcription factors lacks supporting data. Either delete this sentence or clarify that *fru* was identified based on prior evidence or candidate selection.
- Clarify inconsistencies between the stated screen (lines 102–105) and the cited reference (Vleurinck et al., 2016, Ref. 33), which does not report male-specific splicing of *fru*.

12. Male Behavior Description: Please provide a clear list of innate and learned male cooperative behaviors in *Apis*, such as:

- Drone congregation behavior
- Aggregation pheromone production
- Trophallaxis
- Heat production and clustering
- Courtship flight patterns

13. Specific Phrase Suggestions:

- Line 530: “*FruM* facilitates decision processes...”
- Line 540: “*FruM* is required for the emergence of aspects of innate behavior...”
- Line 548: Replace “specifies” with “contributes to”
- Line 620: “We identified a gene that contributes to decision processes...”

In summary, we support the publication of this important work with minor revisions, pending clarification of the above points and the inclusion of appropriate references and discussion. The integration of neuroanatomy, genetics, and behavior is commendable, and with additional detail and balance, this manuscript will be a significant contribution to the field.

Reviewer #3

(Remarks to the Author)

This study contains several important findings that will advance our understanding on the neural basis of cooperative social behavior. Most importantly, results show that the male-specific isoform of the honeybee *fruitless* gene (*FruM*) is involved in the regulation of male behaviors through its neural functions. Although the function of *FruM* on male-specific behaviors has been well characterized in the common fruit fly *Drosophila melanogaster* and its sister species, whether *fruitless* is important for sex-specific behaviors outside dipterans has been ambiguous. Consequently, the genetic origins of sexually dimorphic behaviors in insects have been unclear.

This manuscript is significant beyond honeybee or even insects, because the data suggest that a dedicated transcriptional regulation is a generalized mechanism that specifies sexual dimorphism of neural circuits and behaviors across phyla. In mammals, neurons that express sex hormone receptors (such as estrogen receptor, progesterone receptor, etc) play critical roles in regulating various aspects of reproductive behaviors. Evidence presented in this study generalizes this rule to Hymenoptera, which will draw attentions of wide range of neuroscientists and evolutionary biologists.

Their conclusions are based on high-quality data and careful observations. Successful creation of genome-edited honeybees is a technical accomplishment on its own. The genetic mutation of *fru* provide convincing evidence that *FruM* is required for appropriate expression of male honeybee behaviors. The nervous system-specific yet restricted expression of *FruM* suggests that its main function is the regulation of neural processes. Behaviors of male mutants show intriguing differences from wild type males in social feeding and positions in the nest. Lastly, the functional imaging experiments show that *FruM* influences male behavior not by altering the sensory systems, but likely through central processing of sensory information. Overall, their multidisciplinary characterization of *fru* honeybee mutants has established a new entry point for

investigating the neural basis of sexually dimorphic behavior in social insects.

Although I believe this manuscript should be published promptly, I would like to make a few editorial suggestions.

First, quantitative analysis on the anatomical data can strengthen their conclusions. Movies and images of the *fru*^{P1myrGFP} allele (Fig. 3 and 4, Movie 4-7) are annotated only qualitatively in the text. Some of their claims are not immediately obvious to untrained eyes as green fluorescence seems to be faintly present across the brain. For instance, the fluorescence in the male antennal lobe (including in the macroglomeruli) is unclear, and it is even harder to recognize any difference between the worker bee antennal lobe. Likewise, the stated male-specific labeling of a tract in the lobula and lateral protocerebrum is not obvious because it is unclear where the corresponding tract in female brain is.

My suggestions are as follows:

- 1) show the GFP staining of a wild type bee brain as a negative control. Faint but broad signals of anti-GFP could be due to the background staining. A negative control specimen will help clarify where the allele-specific signals exist.
- 2) Quantify the fluorescence intensity of targeted neuropils where sexually dimorphic labeling were observed, such as ventromedial antennal lobe or the lobula/lateral protocerebrum. Phalloidin staining could serve as a normalizing signal.
- 3) Annotate structures of interest in the movies, so that readers know where to pay attention to. For example, Movie 7 supposedly shows *fru*-expressing antennal nerves only in workers (lines 279-280), but it is impossible to tell where the nerve is. This statement is also confusing since the text says that both males and workers have *fru*-expressing antennal fibers (lines 275-277).

Second, I would like to see the signals of anti-FruM and *fru*^{P1myrGFP} in the posterior end of the brain. Fig. 3d is the only image of the posterior side, and movies appear to stop at around the middle of the brain. Studies in *Drosophila* have shown that posterior medial part of the brain contains cell types that control courtship behavior. It is interesting to address if a cluster of FruM neurons is present in the male honeybee brain or not.

Third, the discussion on the FruM circuitry (line 585-609) can include a brief comparison between the *Drosophila fru* circuit. In *Drosophila*, sexually dimorphic neurons are mostly in the central interneurons but some sensory neurons do express FruM. This seems to be a notable difference in the honeybees (line 279), leaving a question over the genetic mechanism of antennae sexual dimorphism (ref. 14). Classic works on the *Drosophila fru* circuit (Yu et al., *Curr. Biol.* 20:1602-1614 (2010); Cacherio et al., *Curr. Biol.* 20:1589-1601 (2010)) can provide discussion points.

Fourth, the relevance of the discussion over the decision processes (lines 555-583) is unclear. The authors' conclusion that "FruM specifies decision process in the nervous system" (line 548) is based on behavioral phenotypes on the cooperative feeding behaviors (Fig. 5). While other innate behaviors and global response to odors (Fig. 8) are largely unchanged, relatively limited types of odors tested and the lack of glomerular resolution of the imaging experiment leave it possible that response to uncharacterized, behaviorally relevant olfactory cues are indeed affected in *fru* mutants. If this is the case, the effects on cooperative feeding might be caused simply by the loss of relevant sensory inputs, rather than "decision process" as discussed in this subsection. This discussion could be saved for future studies in which the relationship between *fru* and action choice during social interaction in honeybees is better characterized.

However, the overall scientific merit of the manuscript remains very high. I strongly recommend that this work becomes broadly available as soon as points raised above are addressed.

Reviewer #4

(Remarks to the Author)

Version 1:

Reviewer comments:

Reviewer #1

(Remarks to the Author)

The authors have fully addressed all of the suggestions and concerns raised in the earlier review.

Reviewer #2

(Remarks to the Author)

I am satisfied with the reply of the authors and the modifications to the draft.

Reviewer #3

(Remarks to the Author)

The revision of the manuscript by Kohnen et al. addresses my concerns in satisfactory manner. I must admit that the labeling pattern of fruP1-GFP is not necessarily as convincing as the description of the manuscript (for example, the labeling in the core regions of the antennal glomeruli is very faint). However, I acknowledge that the creation of this genetic allele is an accomplishment on its own, which opens an avenue for further characterization of *fru*-expressing neural circuits in honeybees. I am happy with the qualitative description of fruP1-GFP labeling pattern at this stage.

The modification in Result and Discussion sections made the message of the paper more straightforward. However, I find the text sometimes difficult to read due to issues in syntax and punctuation. I don't think a reviewer should serve as a proofreader, but I strongly recommend that the authors use a professional English proofreading service before the manuscript is published. Some errors are relatively easy to read through, but others compromise the clarity of the manuscript, which is a regrettable disservice to this otherwise exciting manuscript.

I managed to note only the following examples. These are by no means comprehensive.

Lines 139-140: "Next, we developed an anti-Fru antibody against a sequence of the BTB domain." "anti-Fru antibody against" is an awkward expression. Something like the following would be better"

"Next, we developed an antibody against a sequence of the Fru BTB domain."

Line 344: "The behavior of each male was behaviorally tracked using..." The expression is redundant. It should be changed to:

"The behavior of each male was tracked using..."

Lines 591-592: "...worker bees within the same age class perform also other tasks..." The location of adverb (also) is incorrect. It should be changed to:

"...worker bees within the same age class also perform other tasks..."

Lines 595-596: "...a large fraction is informed about the food status that can perform task informed decisions just from the local interactions." This is an awkward sentence. The following is a suggestion:

"...a large fraction is informed about the food status that can choose appropriate tasks based only on the local interactions."

Lines 644-646: "In *D. melanogaster* FruM+ expressing OSNs, which detect female pheromones, project into three male-specific enlarged glomeruli 76-78." This sentence is difficult to understand, but I assume authors mean:

"In *D. melanogaster*, FruM-expressing OSNs, which detect female pheromones, project into three glomeruli that are enlarged specifically in males"

Lines 651-653: "These differences revealed that FruM's role diverged in the two species suggesting notable entry points to dissect honeybee confined specifications of the circuitry and cooperative behaviors." The lack of appropriate punctations makes this sentence difficult to understand. It should be changed to something like the following:

"These differences revealed that FruM's role diverged in the two species, suggesting notable entry points to dissect honeybee-confined specifications of the circuitry and cooperative behaviors".

Reviewer #4

(Remarks to the Author)

The responses to the REVIEWERS' COMMENTS

We thank the reviewers for their time and efforts and their helpful comments. They further improved the ms!

Reviewer #1 (Remarks to the Author):

The manuscript by Köhnen et al presents molecular, morphological, and behavioural evidence for the involvement of the fru gene in shaping the feeding behaviour of male honey bees. This study makes a valuable contribution to our understanding of the molecular underpinnings of social behaviour. The ms is well written, clearly presenting the results of elegant experiments and concluding with balanced, insightful discussion. I have only a few minor suggestions, as outlined below:

Introduction

Lines 71-72: This paragraph should not begin with “however,” because this word typically contrasts with a statement just made, and here the contrast is with a point from the previous paragraph (again in line 102).

Response: *We deleted “however” in line 71 and replaced “however” with “thus far” in line 102 (now line 104).*

More importantly, the ms emphasizes the lack of information regarding the molecular mechanisms controlling cooperative behaviours in honey bee males. The behaviours described so far primarily focus on the male-worker interaction where males receive food -a one-way interaction that provides an immediate benefit only to the males. Please also highlight the reciprocal aspect by stressing the indirect benefits that workers (and the colony as a whole) gain from trophallaxis, thereby presenting a clearer view of the studied “cooperative” behaviour.

Response: *Yes, we agree. We now inserted a sentence stating the importance of trophallaxis behaviors for the colony. Line 66: “The trophallaxis behavior plays a central role in the food exchange among colony members, the nutrition of larvae, adult males and queens and the communication about the food status of the colony¹. “*

Line 77: The phrase “Compared with worker bees” is unclear. Could the authors clarify whether males possess the pathway while workers do not? Please provide more details.

Response: *we rephrased the sentence to clarify the content, now line 75: “Male identity that differ from worker bees and queens manifests in male-specific odorant receptor gene expression, male-specific number and anatomy of glomeruli in the primary olfactory processing center (the antennal lobe (AL)), a male-specific smaller mushroom body (MB) harboring 12% fewer neurons and in male-specific larger optical lobes²⁻⁷.”*

Results

Line 115 (and figure 1): Are P1-P3 designated as promoters or as transcriptional start sites? Please clarify their functional roles based on the specific nucleotide sequences or motifs used to define or identify these elements.

Response: we added the requested information (underlined), now line 116: “Our detailed transcript nucleotide sequence analyses revealed that the fru gene in honeybee is transcribed from at least three promoters based on three different transcription start sites”.

We can only state their existence and that the P1 derived transcript is nervous system specific expressed that produce a sex-specific splice product. This P1 derived transcript is involved in the phenotypes which we described in this study.

Additionally, Figure 1c is lacking the female protein representation, which is expected to be strikingly different to that of males. In lines 122-123 include the information on the expected peptide size of the female counterpart.

Response: we are not so sure if this protein is indeed produced. Cellular mechanisms are operating that will inhibit that such small protein is produced from a long transcript with an early translation stop codon (for example the nonsense-mediated mRNA decay mechanism)

To further clarify this issue, we now provided additional information in Fig 1 c and added the expected sizes in line 126 (“34-amino-acid-long protein...”). We now also schematically present in Figure 1 C this small sized peptide. We state in the figure legend line 175: “Note, that if a female protein is produced it will only have 34 amino acids.”

Figure 1e: Are the levels of ef1 α in wt males exactly the same to those in females?

Response: Oh, sorry we used different primer pairs (over the years) to amplify and the ef1 α for the males in Figure 1e was somehow not included. We now added the wt male data of ef1 α in Figure 1e and stated in the figure legend that different primer pairs were used.

We inserted in line 182. (figure legend): “Please note that the ef1 α RT-PCRs in (e) versus (f, g) generated different amplicon sizes because different oligonucleotide primer pairs were used.”

Lines 431-434: I recommend softening this statement, as the type and extent of the morphological analyses conducted do not provide sufficient support for such a strong conclusion.

Response: we softened the statement from

“Malfunction and the malformation of external and internal structures also cannot explain the observed behavioral dysfunction because the external head, antennae, abdomen and leg morphology as well as the reproductive organ anatomy of fruP1⁻ males had a wt phenotype (Fig. 7f). “

Into (now line 444):

“These behavioral dysfunctions also cannot be explained by a malfunction and malformation of external and internal structures at the gross level because the external head, antennae, abdomen and leg morphology as well as the reproductive organ anatomy of fruP1⁻ males had a wt phenotype (Fig. 7f)”

Discussion

Lines 532-536: This idea has been reiterated throughout the manuscript -initially in the Introduction, followed by the Results, and now in the Discussion. I recommend elaborating on it more thoroughly in the Introduction, particularly emphasizing its benefits for the colony as a whole, as previously suggested.

Response: *beside the changes in the introduction (see above) we now provided more arguments concerning the benefits for the colony in the discussion. We inserted (now line 546): "These trophallactic interactions among colony members thereby communicate the food status of the colony¹. From these cooperative behavioral interactions, the colony benefits as a whole thereby increasing its reproductive success.*

Lines 616-619: Has the fru gene been specifically identified as associated with social evolution in any of the 4 papers cited by the authors? If so, do these studies propose any common or divergent mechanisms of action for the gene?

Response: *We were not able to identify the fru gene in these data sets. We better emphasized the focus of this study as compared to the excellent previous studies.*

Line 632:

"In this study, we identified a single gene by functional perturbation that programs initiation and sustainment processes of within-colony cooperative behaviors."

We provided further information in the discussion about how fru function possibly evolved at the molecular level.

Line 652:

"Rather the comparisons suggest that the connectivity and functional properties of Fru^{M+} circuitry diverged in the species that possibly involve the evolution of other downstream controls of Fru protein target genes."

Overall, the ms would benefit from punctuation and spelling editing (e.g. lines 87, 327).

Response: *We carefully edited the entire ms.*

Reviewer #2 (Remarks to the Author):

The manuscript titled "The fru gene specifies male cooperative behaviors in honeybee colonies" presents compelling and novel findings that significantly advance our understanding of the genetic control underlying social behaviors in honeybee colonies. This research provides insightful data that builds upon existing knowledge of social behavior in insects, with a particular focus on the role of genetics in shaping these behaviors. By

exploring the fruitless (*fru*) ortholog in *Apis mellifera*, the authors uncover essential facets of male development and interaction within the hive.

Their investigation reveals that the *fru* gene expresses male-specific products during the late developmental stages of the nervous system, highlighting a critical window in which genetic influences can shape behavioral outcomes. This expression is particularly noteworthy because it suggests that the *fru* gene is active at key points in male maturation, potentially influencing the development of behaviors necessary for reproductive success and social integration within the colony. These findings align with observations in *Drosophila* and other dipterans, where the *fru* gene has also been implicated in regulating mating behaviors, demonstrating a conserved mechanism across different species.

However, the manuscript goes beyond mere correlation and establishes a direct link between the male-specific *fru* products and various behavioral paradigms within honeybee colonies. Notably, these male-enriched *fru* products are shown to be essential not only for modulating male courting behaviors but also for regulating social feeding behaviors that are vital for the colony's overall function. This dual role of the *fru* gene suggests a complex interplay between mating and cooperative social interactions, providing a deeper understanding of how evolutionary pressures may have shaped these behaviors to promote colony survival.

The research illustrates how the expression of male-specific *fru* products is crucial for drones to exhibit typical behaviors such as normal rates of begging for food, the duration of trophallaxis—an important process in which food is exchanged among colony members—and the dynamics of interactions with worker bees. These interactions are critical, as they help to maintain the nutritional health of the colony and ensure a cohesive social structure. The results underscore the fundamental importance of these behaviors in the daily functioning of the hive, emphasizing the role of males beyond reproduction.

A particularly striking aspect of the study is the documentation of the consequences of manipulating *fru* gene expression. The authors provide compelling evidence that drones with a specific disruption in the expression of the *fruM* isoform exhibit a dramatic two- to threefold decrease in the execution of essential social behaviors, such as begging and trophallaxis. This finding provides robust support for the hypothesis that the *fru* gene is a key regulator of male-specific behaviors that facilitate not only individual reproductive opportunities but also the overall health and efficiency of the colony.

In summary, this manuscript significantly enriches our understanding of the genetic underpinnings of social behavior in honeybees, illuminating the intricate connections between genetics, behavior, and social structure within colonies. The implications of this research extend beyond honeybees, potentially informing studies on behavioral genetics in other social insects and adding depth to our comprehension of the evolutionary biology of complex social systems. By elucidating the multifaceted role of the *fru* gene in male-specific behaviors, this work sets the stage for future investigations into the genetic and molecular mechanisms that orchestrate social interactions in eusocial species.

To further improve the manuscript, we propose minor revisions and request the authors to address the following points:

Previous studies in *Drosophila* have demonstrated that male expression of the fruitless gene is not only necessary for male courtship behaviors but is also sufficient to induce male-like behaviors in otherwise normal females (Demir & Dickson, 2005, *Cell*). This significant finding led to the conclusion that the *fru* gene plays a central role in specifying various aspects of complex innate behaviors. Given this context, we are interested in understanding why the authors did not incorporate gain-of-function experiments—specifically, the expression of *fruM* in the brains of female workers through methods such as deleting the female-specific exon—to rigorously test whether *fruM* is sufficient to activate male-like cooperative behaviors in otherwise normal female workers.

Response: *Such an experiment is not the focus of this study, which is clearly stated in the introduction (see below). We wanted to identify genes that are involved and required for the social within colony behaviors. We do not state anything about a master regulator function. The experiment conducted by Demir and Dickson had the goal to answer the question if thr*

fru gene is a master regulator of behavioral specification, which requires an essential and sufficient analysis. Such master regulators have been identified in developmental programs. They were key to understand the organization of the body plan formation. And it was an open question whether such master regulator also exists for behavioral specifications.

Taken from the introduction defining the goal of this study.

*“Thus far, we do not know which genes took on the roles in hardwiring the male’s cooperative behaviors in the nervous system. We thus screened for male-specific spliced transcripts in the nervous system⁸ and found that, in addition to previously known *glu* and *dsx*, only the fruitless (*fru*) TF gene has male-specifically spliced transcripts. Because this Fru BTB zinc finger transcription factor is expressed only in the male nervous system and its homolog in *Drosophila melanogaster* is involved in hardwiring sexual behaviors^{9, 10}, we set out to examine the function of this gene product in manifesting the male’s cooperative behaviors in honeybee colonies.”*

This omission is particularly intriguing as it presents an opportunity to explore the sufficiency of fruM in reshaping behavioral phenotypes traditionally associated with male *Drosophila*. By examining the potential to induce male-like behaviors in females, the results of such experiments could significantly enhance our understanding of how fruM integrates with existing neural circuits involved in cooperative behavior. Furthermore, additional insights could clarify the genetic and neurobiological underpinnings of sex-specific behaviors, potentially revealing broader implications for understanding the evolution of behavioral traits across species.

Response: *The goal of this study is to identify a transcription factor which is essential for the specification and expression of the male specific behavior in the colony. Hence, this what we presented in this manuscript. No doubt this is an interesting question which deserve an extra study as the one from Demir and Dickson.*

General Comments and Suggestions for Revision:

1. Title Revision: We propose changing the title from “The fru gene specifies male cooperative behaviors in honeybee colonies” to “The fru gene contributes to male cooperative behaviors in honeybee colonies” to better reflect the data, which demonstrate necessity but not sufficiency.

Response: *Thank you for your efforts to improve the manuscript. However, your title suggestion does not cover what we want to present as the focus of this study. Our study revealed that the fru gene is required or essential for the specification of some aspects of social within colony behaviors. The title is a short summary of that. The word specification is actually used to reflect this partial effect. Not all behavior is “controlled” by this gene. And specification is used to reflect that the gene specifies during development the capacity of the adult male to express behavior. Contribution does not cover the latter mechanism. Similarly, imagine a development study in which someone would state in the title “The X gene contribute to antennae”. The process is missing.*

2. Missing Gain-of-Function Evidence: The authors provide strong loss-of-function data using CRISPR/Cas9. However, gain-of-function experiments (e.g., ectopic expression of fruM in females by deleting the female-specific exon) would be crucial to test sufficiency. We request that the authors comment on why this was not attempted.

Response: *As outlined above, a gain of function experiment is not the focus of this study. We are interested to find transcription factors that are essential for the specification of the within colony behaviors.*

3. Reference to Previous Work:

- The authors should cite and compare their findings on fru gene structure and splicing isoforms in *Apis mellifera* with *Nasonia vitripennis* (Bertossa et al., 2009, Mol Biol Evol. PMID: 19349644).
- Additionally, they should reference Siwicki & Kravitz (2009, Curr Opin Neurobiol) for background on fru and dsx in *Drosophila* behavior.
- Consider also Gailey et al. (2006, PNAS, PMID: 16319090) regarding the evolutionary conservation of fruitless function.

Response: *thank you for letting us know about other publications which we may have missed. After re-evaluating those and comparing those with the ones we have cited and most importantly, the focus of our study (please take into account that we have a restriction of 70 citations) we finally came to the conclusion that we prefer the publications cited. Here the reasons. Our evolutionary analysis did not cover “gene structure” as a phenotype. As you know, such feature can evolve fast. Instead we included those studies which showed evidence that fru gene has a conserved function in specifying sex-specific behaviors, which is the very focus of this study. We cited here the relevant publications. We also examine cooperative behavior and not general social behavior. Interesting, some authors refer to sexual behavior as social behavior. The latter is an interesting topic on its own.*

To further improve the ms we better specified what we evolutionary compared here. We inserted (underlined) line 635 “The fru gene also plays a role in hardwiring mating behaviors in honeybees as in other insects. They all use the same mechanism of sex-specific fruP1 transcript splice regulation which suggests a deep conservation of fru’s role in specifying sexual behaviors”¹¹⁻¹⁴. “

4. Clarification on Expression and Antibody Specificity: The manuscript notes that the antibody detects both male-specific FruM and a common isoform. The authors should clarify if the common isoform is undetectable or weakly expressed in both sexes, and whether P2/P3 promoters are active in the brain.

Response: *Our focus is to examine the male specific program and thus we are interested in the male specific expressed Fru protein. Our immunostainings used our developed Anti-Fru antibody. They showed a male-specific labeling pattern suggesting that these labels derived from P1 transcript. We functionally tested that by deleting P1 promoter region. Indeed, this Anti-Fru labels were lost in the knockouts. Hence, we conclude that these male-specific specific labels are derived from P1 transcript expression. This also suggests specificity of our AB. Whether there is a weak P2/P3 derived expression in both sexes that are near the detection level using our Anti-Fru antibody is hard to tell. Anyway, the common fru proteins which are common to both sexes cannot specify a male-specific behavior. And the male-specific behavior is the focus of this study.*

We improved the ms in that respect. We added in line 144: “...suggesting that these male-specific labels derived from P1 and not P2 or P3 transcripts.”

Line 152: “These results suggest that the male-specific FruM protein is expressed from the male fruM transcripts, that the anti-Fru antibody labels Fru protein expression and that the fruP1- mutant is a loss-of-function mutation of fruM.”

5. Discussion of Additional Genetic Contributors: While the paper attributes behavioral phenotypes to fruM loss, the observed partial phenotypes suggest other genes likely act in parallel. The authors should discuss candidate genes or regulatory pathways that might also contribute.

Response: *We think that this is too speculative to highlight possible other factors given 12,000 genes of the honeybee genome and to discuss whether they act in parallel.*

6. Potential Role of dsx: Drosophila studies show that dsxM and fruM act in concert to establish male courtship behavior. The authors should comment on why they did not knock down dsxM alone or in combination with fruM.

Response: These are very interesting experiments which we would love to do in the future. Again, our study has a focus, which we clearly stated in the introduction, which is the following:

“Because this Fru BTB zinc finger transcription factor is expressed only in the male nervous system and its homolog in Drosophila melanogaster is involved in hardwiring sexual behaviors^{9, 10}, we set out to examine the function of this gene product in manifesting the male’s cooperative behaviors in honeybee colonies.”

7. Clarification of RNAi and Developmental Timing:

- The authors should explain why fruM mRNA was detected in fem RNAi-treated female larvae alongside substantial female-specific fru transcripts.

Response: *We improved this section and added further information. We showed by performing this experiment that the P1 fru gene is regulated by the sex determination cascade. The effect is partial as suggested because knockdown is only temporally induced (see^{15, 16}).*

- They should also provide the developmental timeline (egg to L4 larva), duration of RNAi effect, and siRNA sequences used.

Response: *The details of the experimental procedure on fem knockdown has been described in previous studies, which can be found here^{15, 16}. The Gempe et al paper 2009 showed that an early downregulation of the transcript via RNAi in the embryo resume at larval stage 5 (see Figure 4 in Gempe et al.). At this stage we see resumed transcripts in fem RNAi females (maintained via a positive feedback loop as also shown in the Gempe et al study) that induce only a partial switch at the level of the fru gene (this study).*

The changes we have made to improve this section: We added in line 129 the following information (underlined):

“The fem RNAi-treated females produced male-specific fruP1 transcripts but also female-specific transcripts at larval stage 4 (Fig. 1e; Supplementary Table 2) suggesting a partial shift toward male splicing in response to temporally restricted fem knockdown which we induced in the embryo¹⁵”

We added in the material section (underlined).

“Female eggs were injected with fem siRNAs and larvae were reared to 4th instar (L4) inducing a temporally restricted knockdown as described¹⁵”

8. RT-PCR Band Size Discrepancy: In Fig. 1 panels e and f, fruM and ef1a bands differ in size. The authors should clarify this, providing primer sequences and amplicon locations.

Response: Sorry for this confusion. We used different primer pairs for the amplification in figure 1 e versus f/g. This has historical reasons. We changed the primer pair over the years to smaller amplicons for various reasons. We now added the primer pairs and stated that these are different primer pairs. The different amplicons do not compromise the conclusion.

We improve the manuscript. We added a sentence in Figure 1 legend line 182: "Please note that the ef1 α RT-PCRs in (e) versus (f, g) generated different amplicon sizes because different oligonucleotide primer pairs were used."

9. Clarify Methodological Details:

- sgRNA design and efficiency for CRISPR deletions should be reported.

Response: The sequences and how we designed the sequences have been reported in the material and method section and the relevant citation. The deletion efficiency is not added in line 12 : "The deletion rate was 14%."

- Clarify ambiguous phrases such as "Numbers: larval and pupal stages,"

Response: we improved thus phrase. "Numbers above gel lanes represent the larval and pupal stages."

and rephrase statements like "it is unclear where FruM is expressed" (when prior data showed expression).

Response: Hm. We refer to where it is expressed not if it is expressed in the brain. This is correct.: "As it is unclear where Fru^M is expressed in the male brain..."

10. Gene-Centric Language: Several statements over-attribute behavioral control directly to a single gene. For example:

Response: *We hopefully clarified this point by now given the above explanation. We together agree that a gene cannot control such behavior. We already have devoted a whole chapter on how such gene can specify the behaviors. Please see line 482 on for further information. "How can the FruM protein specify these male behaviors? We propose that the FruM protein specifies connectivity or functional features in the nervous system that determine behavioral control. To understand the role of the FruM protein in the programming of behaviors, we examined chemosensory receptor abundance, neuropil anatomy and odor processing in fruP1- males. ... "*

Our work asks the question how the behavioral capacity is established through genetic specification. Tthe word "contribution" is misleading as it provides in our understanding no link to an underlying process. (For example: gene X contribute to antennae is not a helpful description of a developmental process).

- "FruM specifies decision processes..." should be changed to "FruM facilitates decision processes..."

Response: We changed into “*Fru^M specifies imitation and sustainment processes in the nervous system that regulate motor program elements of cooperative behavior*”

- “...programs decision processes...” should be softened to “contributes to the neural basis of...”

Response: We changed as above: “...programs initiation and sustainment processes of within-colony cooperative behaviors.”

- These revisions would avoid an overly reductionist interpretation and better reflect the multifactorial nature of behavior.

11. Abstract and Introduction Revisions:

- Line 35: Replace “whether” with “how” — “but how cooperative behaviors require...”

Response: we ask whether there is a dedicated program for the cooperative behavior. This is correct. This is the question we ask (see above. Also other researcher asked that questions for example in the case of the fru gene.

- Lines 39–40: The claim about screening transcription factors lacks supporting data. Either delete this sentence or clarify that fru was identified based on prior evidence or candidate selection.
- Clarify inconsistencies between the stated screen (lines 102–105) and the cited reference (Vleurinck et al., 2016, Ref. 33), which does not report male-specific splicing of fru.

Response: This is all correctly stated. We understand that we need to provide further information. Vleurinck et al., 2016, Ref. 33 describes the genome-wide search for sex-specific spliced transcripts using the Spanki software program and the generated RNAseq data set of the pupal brain (which is in detail described in the above citation). We now provide a detailed list of genes that we identified as sex-specific spliced and highlight those that were transcription factors (TFs) (now supplementary data table 1).

Changes: we added a supplementary table with the list of sex-specific spliced genes described in Ref. 33 and marked the subset of genes that encode transcription factor protein (line 106).

The title of the table: “List of sex-specific spliced genes in the pupal brain and the subset of genes that encode a transcription factor “

12. Male Behavior Description: Please provide a clear list of innate and learned male cooperative behaviors in Apis, such as:

- Drone congregation behavior
- Aggregation pheromone production
- Trophallaxis
- Heat production and clustering
- Courtship flight patterns

Response: *Thanks for your suggestion. However, this information is not the focus of our research question. Key publications that we cited and that describe our within colony behavioral phenotypes are those ^{1, 17}.*

13. Specific Phrase Suggestions:

- Line 530: “FruM facilitates decision processes...”
- Line 540: “FruM is required for the emergence of aspects of innate behavior...”
- Line 548: Replace “specifies” with “contributes to”
- Line 620: “We identified a gene that contributes to decision processes...”

Response: *We suggest (see above) above that the wording “contribution” is not what we want to state here. It is hard to imagine how a gene can contribute to a behavior. It is the specification of the nervous system. We hope that the reviewer is by now convinced that this at least an alternative. It is about the specification of the nervous system. We replaced “decision” with “initiation and sustainment”.*

In summary, we support the publication of this important work with minor revisions, pending clarification of the above points and the inclusion of appropriate references and discussion.

Responses: *We added relevant citations in places, added more information and sharpened our discussion in the places indicated.*

The integration of neuroanatomy, genetics, and behavior is commendable, and with additional detail and balance, this manuscript will be a significant contribution to the field.

Reviewer #3 (Remarks to the Author):

This study contains several important findings that will advance our understanding on the neural basis of cooperative social behavior. Most importantly, results show that the male-specific isoform of the honeybee *fruitless* gene (FruM) is involved in the regulation of male behaviors through its neural functions. Although the function of FruM on male-specific behaviors has been well characterized in the common fruit fly *Drosophila melanogaster* and its sister species, whether *fruitless* is important for sex-specific behaviors outside dipterans has been ambiguous. Consequently, the genetic origins of sexually dimorphic behaviors in insects have been unclear.

This manuscript is significant beyond honeybee or even insects, because the data suggest that a dedicated transcriptional regulation is a generalized mechanism that specifies sexual dimorphism of neural circuits and behaviors across phyla. In mammals, neurons that express sex hormone receptors (such as estrogen receptor, progesterone receptor, etc) play critical roles in regulating various aspects of reproductive behaviors. Evidence presented in this study generalizes this rule to Hymenopterans, which will draw attentions of wide range of neuroscientists and evolutionary biologists.

Their conclusions are based on high-quality data and careful observations. Successful creation of genome-edited honeybees is a technical accomplishment on its own. The genetic mutation of *fru* provide convincing evidence that FruM is required for appropriate expression

of male honeybee behaviors. The nervous system-specific yet restricted expression of FruM suggests that its main function is the regulation of neural processes. Behaviors of male mutants show intriguing differences from wild type males in social feeding and positions in the nest. Lastly, the functional imaging experiments show that FruM influences male behavior not by altering the sensory systems, but likely through central processing of sensory information. Overall, their multidisciplinary characterization of *fru* honeybee mutants has established a new entry point for investigating the neural basis of sexually dimorphic behavior in social insects.

Although I believe this manuscript should be published promptly, I would like to make a few editorial suggestions.

First, quantitative analysis on the anatomical data can strengthen their conclusions. Movies and images of the *fru*^{P1myrGFP} allele (Fig. 3 and 4, Movie 4-7) are annotated only qualitatively in the text. Some of their claims are not immediately obvious to untrained eyes as green fluorescence seems to be faintly present across the brain. For instance, the fluorescence in the male antennal lobe (including in the macroglomeruli) is unclear, and it is even harder to recognize any difference between the worker bee antennal lobe. Likewise, the stated male-specific labeling of a tract in the lobula and lateral protocerebrum is not obvious because it is unclear where the corresponding tract in female brain is.

My suggestions are as follows:

1) show the GFP staining of a wild type bee brain as a negative control. Faint but broad signals of anti-GFP could be due to the background staining. A negative control specimen will help clarify where the allele-specific signals exist.

Response: *This is a very helpful idea. The changes we did. We now included these controls as a movie (Supplementary movie 12 and 13) and added this male negative control directly in Figure 4 in order to show the specificity also of the faint signals. As you can see the anti-GFP staining generated a nearly black background in the negative control (see below). We also reorganized Figure 4 to better present the sex-dimorphic structures. We added the relevant information to the movie and figure 4 legend, line 312: "Anti-GFP staining of wt males were used as negative control."*

2) Quantify the fluorescence intensity of targeted neuropils where sexually dimorphic labeling were observed, such as ventromedial antennal lobe or the lobula/lateral protocerebrum. Phalloidin staining could serve as a normalizing signal.

Response: *This is our first step of labeling such structure in bees and social insects in general. It is an initial step and at this stage our goal is to qualitatively describe differences between the sexes. The negative control establishes a nearly black background confirming that the labels were specific. We did the following: (i) improved the presentation of the sex dimorphic structures in the figure 4 to support our claim, (ii) presented the negative control in Figure 4.*

Our current analysis is not at high spatial resolution. The goal at this step is to only broadly describes the pattern. In a next and future step, we would like to quantify differences at high resolution. However, this requires more genetic engineering of the bees than the current knockin. We would like to employ amplifications of GFP protein expressing using a genetic binary system such as a Gal4/UAS or Q. Hence, at this first step, we would like to stay with qualitative and structural differences.

The changes we have made to improve the ms:

We included a negative control and improved Figure 4. We changed Figure 4 a to d and improved the arrow and arrowhead presentation and description in order to better mark the structures. We now also marked these structures in supplementary movies 6, 7, 11 and 13. We rewrote the paragraph in the main ms to more clearly describe the differences.

Line 285 on: “The antennal nerve is labeled in both male and female worker bees (arrow, Fig. 4a, Supplementary Movie 6, 11 and 13). In workers, a small number (approximately a dozen) of glomeruli on the ventro-medial side of the AL had *myrGFP^{P1+}* labels both in the core and cortex (arrow, Fig. 4b), whereas in males, *Fru^{M+}* labels were consistently restricted to the core (arrowhead, Fig. 4b). This female-specific labeling of the cortex suggests that OSNs in the antennal nerve are *Fru^{M+}* negative in males and *myrGFP^{P1+}* positive in females. This is supported by a more restricted labeling of the nerve in males than in females (arrows, Fig. 4a: note, that the antennal nerve contains approximately 4 times more neurons in males than in female workers³). Hence, sensory neurons from the antennal nerve that bypass the antennal lobe (AL) and glomeruli on the ventral side (antennal tracts T5 & T6) towards the antennal mechanosensory and motor center (AMMC) and potentially the subesophageal zone (SEZ) were labeled in males but also in female worker bees. The set of worker-specific labeled glomeruli may correspond to the T3b cluster and could be involved in the processing of information derived from cuticular hydrocarbon perception^{3, 18, 19}. *Fru^{M+}* circuitry is present in enlarged macroglomeruli in the core, which are male-specific structures that are absent in workers (MG1, Fig. 4b, Supplementary Movie 6, 11 and 13). The *Fru^{M+}* circuitry of the male visual system has a male-specific tract (large arrow, Fig. 4c, Supplementary Movie 7) that projects from the lobula (LO) to the posterior lateral protocerebrum (PLP) but shares two other tracts with the *myrGFP^{P1+}* circuitry of worker bees in the same area (small arrows, Fig. 4c). In the central complex, a few labeled *myrGFP^{P1+}* fibers were found in the noduli (NO) of female worker bees, but no *Fru^{M+}* cells were found in males (arrows, Fig. 4d, Supplementary Movie 6, 11 and 13)²⁰. We conclude from these comparisons that the *Fru^{M+}* circuit has a male-specific identity that manifests at the gross level in the anatomy of the antennal nerve, macroglomeruli, and innervation of the core/cortex of the glomeruli and optic tracts.”

3) Annotate structures of interest in the movies, so that readers know where to pay attention to. For example, Movie 7 supposedly shows *fru*-expressing antennal nerves only in workers (lines 279-280), but it is impossible to tell where the nerve is. This statement is also confusing since the text says that both males and workers have *fru*-expressing antennal fibers (lines 275-277).

Response: Yes, they were not easy to find sorry. We now carefully labeled these structures in the movies and explained these labels in the movie descriptions (now movies 6, 7, 11 and 13). We rewrote this entire paragraph (see above) to better explain and delete confusing statements.

Second, I would like to see the signals of anti-FruM and *fru^{P1myrGFP}* in the posterior end of the brain. Fig. 3d is the only image of the posterior side, and movies appear to stop at around the middle of the brain. Studies in *Drosophila* have shown that posterior medial part of the brain contains cell types that control courtship behavior. It is interesting to address if a cluster of FruM neurons is present in the male honeybee brain or not.

Response: It appears that we needed a 2nd round of stacks with higher laser intensity to make these posterior parts visible. Sorry for not presenting them. We now present these stacks as extra movies (Supplementary Movie 7 and 9; posterior region). We were not able to characterize the cluster of FruM neurons, the ones the reviewer mentioned. We now stated about these additional movies in the main text.

Third, the discussion on the FruM circuitry (line 585-609) can include a brief comparison between the *Drosophila fru* circuit. In *Drosophila*, sexually dimorphic neurons are mostly in the central interneurons but some sensory neurons do express FruM. This seems to be a notable difference in the honeybees (line 279), leaving a question over the genetic mechanism of antennae sexual dimorphism (ref. 14). Classic works on the *Drosophila fru* circuit (Yu et al., Curr. Biol. 20:1602-1614 (2010); Cachero et al., Curr. Biol. 20:1589-1601 (2010)) can provide discussion points.

Response: We now added a brief comparison to *D. melanogaster* in the last paragraph of the discussion and cited the relevant studies. The reviewer is correct: there is a notable difference to the fruit fly: In honeybee males Fru^M is not expressed in the OSNs that enter the glomeruli (see explanation and Fig. 4 b). Fru^M is also not required for the sexual dimorphic development of the male-specific enlarged macroglomeruli (Fig. 8a) and for the processing of queen's pheromone information at the level of the macroglomeruli MG 2 (Fig. 8b to d).

Line 640 we now wrote: “. In *D. melanogaster* Fru^{M+} expressing OSNs, which detect female pheromones, project into three male-specific enlarged glomeruli²¹⁻²³. The formation of these sexual dimorphic glomeruli depends in the fruit fly on the male-specific activity of Fru protein²¹⁻²³. To the contrary, Fru^M in honeybee males is not expressed in the OSNs that enter the glomeruli (Fig. 4a, b), is not required for the sexual dimorphic development of the male-specific enlarged macroglomeruli (Fig. 8a) and for the processing of queen's pheromone information at the level of the macroglomeruli MG 2 (Fig. 8b to d).”

Fourth, the relevance of the discussion over the decision processes (lines 555-583) is unclear. The authors' conclusion that "FruM specifies decision process in the nervous system" (line 548) is based on behavioral phenotypes on the cooperative feeding behaviors (Fig. 5). While other innate behaviors and global response to odors (Fig. 8) are largely unchanged, relatively limited types of odors tested and the lack of glomerular resolution of the imaging experiment leave it possible that response to uncharacterized, behaviorally relevant olfactory cues are indeed affected in *fru* mutants. If this is the case, the effects on cooperative feeding might be caused simply by the loss of relevant sensory inputs, rather than “decision process” as discussed in this subsection. This discussion could be saved for future studies in which the relationship between *fru* and action choice during social interaction in honeybees is better characterized.

Response: We agree that this could be misleading. Our intension was to anchor the discussion around the behavioral phenotype in some places. The rate/duration impairment is an interesting phenotype to understand the innate social task organization in the colony. We now deleted the words "decision process" and replaced it with the words "initiation and duration" (or related). Accordingly, we rewrote the sentences to more clearly distinguish between the behavioral phenotype and the control driven by the nervous system. This required rewritings of the discussion section.

This includes:

Line 553: “Specifically, we found that *fruP1* transcripts are required for the rate of approaching, begging and trophallaxis behaviors, suggesting that the initiation of three sequences of the social feeding behavior are innately programmed by Fru^M. The duration of trophallaxis behavior also requires the Fru^M protein, suggesting that there is a neural basis for a sustainment process that controls the duration of social feeding. The choice of whether males contact pollen food sources in the cells also requires *fruP1* transcripts, indicating that Fru^M protein programs repression of the cell entering for the pollen stimulus. This repression of pollen interacting behaviors reduce social conflicts among colony members over the collectively used food stores. Overall, we showed that Fru^M specifies imitation and

sustainment processes in the nervous system that regulate motor program elements of cooperative behaviors.”

Line 574: “This specification of the rate and duration aspects has profound consequence in the colony as it innately scales the bee's participation in the social feeding task.

We now better structured the discussion and more clearly stated the two possibilities where the rate impairment comes from. Line 600: “The *Fru^M* protein is required for initiation and sustainment of social feeding behaviors and is spatially restricted expressed in the nervous system suggesting that the *Fru^M*-expressing circuitry is involved in controlling these aspects of behaviors. This *Fru^{M+}* circuitry is involved in processing olfactory, gustatory and mechanosensory information, suggesting a possible role of different sensory modalities in the control of these within-colony behaviors^{3, 7, 18, 19, 24, 25}. *Fru^{M+}* circuitries are also found in higher-order processing centers—the mushroom bodies—indicating that the circuitry is possibly involved in evaluating and integrating sensory information for behavioral decision processes²⁴⁻²⁸.”

However, the overall scientific merit of the manuscript remains very high. I strongly recommend that this work becomes broadly available as soon as points raised above are addressed.

Reviewer #4 (Remarks to the Author):

References:

1. Crailsheim K. Trophallactic interactions in the adult honeybee (*Apis mellifera* L.). *Apidologie* **29**, 97-112 (1998).
2. Jain R, Brockmann A. Sex-specific molecular specialization and activity rhythm-dependent gene expression in honey bee antennae. *J Exp Biol* **223**, (2020).
3. Mariette J, Carcaud J, Sandoz JC. The neuroethology of olfactory sex communication in the honeybee *Apis mellifera* L. *Cell Tissue Res* **383**, 177-194 (2021).
4. Wanner KW, Nichols AS, Walden KK, Brockmann A, Luetje CW, Robertson HM. A honey bee odorant receptor for the queen substance 9-oxo-2-decenoic acid. *Proceedings of the National Academy of Sciences of the United States of America* **104**, 14383-14388 (2007).
5. Witthöft W. Absolute Anzahl und Verteilung der Zellen im Hirn der Honigbiene. *Zeitschrift für Morphologie der Tiere* **61**, 160-184 (1967).
6. Marco Antonio DS, Hartfelder K. Toward an Understanding of divergent compound eye development in drones and workers of the honeybee (*Apis mellifera* L.): A correlative analysis of morphology and gene expression. *J Exp Zool B Mol Dev Evol* **328**, 139-156 (2017).

7. Sandoz JC. Odour-evoked responses to queen pheromone components and to plant odours using optical imaging in the antennal lobe of the honey bee drone *Apis mellifera* L. *J Exp Biol* **209**, 3587-3598 (2006).
8. Vleurinck C, Raub S, Sturgill D, Oliver B, Beye M. Linking genes and brain development of honeybee workers: a whole-transcriptome approach. *PLoS One* **11**, e0157980 (2016).
9. Rings A, Goodwin SF. To court or not to court - a multimodal sensory decision in *Drosophila* males. *Curr Opin Insect Sci* **35**, 48-53 (2019).
10. Kimura K, Hachiya T, Koganezawa M, Tazawa T, Yamamoto D. *Fruitless* and *doublesex* coordinate to generate male-specific neurons that can initiate courtship. *Neuron* **59**, 759-769 (2008).
11. Basrur NS, *et al.* *Fruitless* mutant male mosquitoes gain attraction to human odor. *Elife* **9**, (2020).
12. Ryner LC, *et al.* Control of male sexual behavior and sexual orientation in *Drosophila* by the *fruitless* gene. *Cell* **87**, 1079-1089 (1996).
13. Ueno M, Nakata M, Kaneko Y, Iwami M, Takayanagi-Kiya S, Kiya T. *fruitless* is sex-differentially spliced and is important for the courtship behavior and development of silkworm *Bombyx mori*. *Insect Biochem Mol Biol* **159**, 103989 (2023).
14. Meier N, Kappeli SC, Hediger Niessen M, Billeter JC, Goodwin SF, Bopp D. Genetic control of courtship behavior in the housefly: evidence for a conserved bifurcation of the sex-determining pathway. *PLoS One* **8**, e62476 (2013).
15. Gempe T, Hasselmann M, Schiott M, Hause G, Otte M, Beye M. Sex determination in honeybees: two separate mechanisms induce and maintain the female pathway. *PLoS Biol* **7**, e1000222 (2009).
16. Hasselmann M, Gempe T, Schiott M, Nunes-Silva CG, Otte M, Beye M. Evidence for the evolutionary nascence of a novel sex determination pathway in honeybees. *Nature* **454**, 519-522 (2008).
17. Ohtani T. Behavior repertoire of adult drone honeybee within observation hives. *Jour Fac Sci Hokkaido Univ SerVI, Zool* **19**, 706-721 (1974).
18. Galizia CG, Sachse S, Rappert A, Menzel R. The glomerular code for odor representation is species specific in the honeybee *Apis mellifera*. *Nat Neurosci* **2**, 473-478 (1999).
19. Paoli M, Galizia GC. Olfactory coding in honeybees. *Cell Tissue Res* **383**, 35-58 (2021).
20. Kaiser A, *et al.* A three-dimensional atlas of the honeybee central complex, associated neuropils and peptidergic layers of the central body. *J Comp Neurol* **530**, 2416-2438 (2022).
21. Stockinger P, Kvitsiani D, Rotkopf S, Tirián L, Dickson BJ. Neural circuitry that governs male courtship behavior. *Cell* **121**, 795-807 (2005).
22. Cachero S, Ostrovsky AD, Yu JY, Dickson BJ, Jefferis GS. Sexual dimorphism in the fly brain. *Curr Biol* **20**, 1589-1601 (2010).
23. Yu JY, Kanai MI, Demir E, Jefferis GS, Dickson BJ. Cellular organization of the neural circuit that drives *Drosophila* courtship behavior. *Curr Biol* **20**, 1602-1614 (2010).
24. Habenstein J, Grubel K, Pfeiffer K, Rössler W. 3D atlas of cerebral neuropils with previously unknown demarcations in the honey bee brain. *J Comp Neurol* **531**, 1163-1183 (2023).
25. Strausfeld NJ. Organization of the honey bee mushroom body: representation of the calyx within the vertical and gamma lobes. *J Comp Neurol* **450**, 4-33 (2002).

26. Arican C, Schmitt FJ, Rossler W, Strube-Bloss MF, Nawrot MP. The mushroom body output encodes behavioral decision during sensory-motor transformation. *Curr Biol* **33**, 4217-4224 e4214 (2023).
27. Strube-Bloss MF, Rossler W. Multimodal integration and stimulus categorization in putative mushroom body output neurons of the honeybee. *R Soc Open Sci* **5**, 171785 (2018).
28. Noyes NC, Davis RL. Innate and learned odor-guided behaviors utilize distinct molecular signaling pathways in a shared dopaminergic circuit. *Cell Reports* **42**, 112026 (2023).

Round 2: Responses to REVIEWERS' COMMENTS

Reviewer #1 (Remarks to the Author):

The authors have fully addressed all of the suggestions and concerns raised in the earlier review.

Reviewer #2 (Remarks to the Author):

I am satisfied with the reply of the authors and the modifications to the draft.

Reviewer #3 (Remarks to the Author):

The revision of the manuscript by Kohnen et al. addresses my concerns in satisfactory manner. I must admit that the labeling pattern of fruP1-GFP is not necessarily as convincing as the description of the manuscript (for example, the labeling in the core regions of the antennal glomeruli is very faint). However, I acknowledge that the creation of this genetic allele is an accomplishment on its own, which opens an avenue for further characterization of *fru*-expressing neural circuits in honeybees. I am happy with the qualitative description of fruP1-GFP labeling pattern at this stage.

The modification in Result and Discussion sections made the message of the paper more straightforward. However, I find the text sometimes difficult to read due to issues in syntax and punctuation. I don't think a reviewer should serve as a proofreader, but I strongly recommend that the authors use a professional English proofreading service before the manuscript is published. Some errors are relatively easy to read through, but others compromise the clarity of the manuscript, which is a regrettable disservice to this otherwise exciting manuscript.

I managed to note only the following examples. These are by no means comprehensive.

Response: Indeed, the reviewer is correct: there were still several places to improve the ms. We did another round of careful editing. We followed the suggestions below.

Lines 139-140: "Next, we developed an anti-Fru antibody against a sequence of the BTB domain." "anti-Fru antibody against" is an awkward expression. Something like the following would be better"

"Next, we developed an antibody against a sequence of the Fru BTB domain."

Response: We deleted "anti-Fru"

Line 344: "The behavior of each male was behaviorally tracked using..." The expression is redundant. It should be changed to:

"The behavior of each male was tracked using..."

Response: we followed these changes

Lines 591-592: "...worker bees within the same age class perform also other tasks..." The location of adverb (also) is incorrect. It should be changed to:

"...worker bees within the same age class also perform other tasks..."

Response: Oh sorry, thanks: we changed it.

Lines 595-596: "...a large fraction is informed about the food status that can perform task informed decisions just from the local interactions." This is an awkward sentence. The following is a suggestion:

"...a large fraction is informed about the food status that can choose appropriate tasks based only on the local interactions."

Response: OK: we improved the entire paragraph. We realized that this paragraph was still hard to understand.

We changed into: "If a large fraction of colony members participates in social feeding, a large fraction is informed about the food status. As a consequence, a large fraction of colony members can adjust their task decisions to colony need⁶⁹, when encountering task-related stimuli."

Lines 644-646: "In *D. melanogaster* FruM+ expressing OSNs, which detect female pheromones, project into three male-specific enlarged glomeruli 76-78." This sentence is difficult to understand, but I assume authors mean:

"In *D. melanogaster*, FruM-expressing OSNs, which detect female pheromones, project into three glomeruli that are enlarged specifically in males"

Response: thanks for your suggestion, which we used this wording to improve the ms.

Lines 651-653: "These differences revealed that FruM's role diverged in the two species suggesting notable entry points to dissect honeybee confined specifications of the circuitry and cooperative behaviors." The lack of appropriate punctations makes this sentence difficult to understand. It should be changed to something like the following:

"These differences revealed that FruM's role diverged in the two species, suggesting notable entry points to dissect honeybee-confined specifications of the circuitry and cooperative behaviors".

Response: many thanks: we followed this suggestion.

Reviewer #4 (Remarks to the Author):
